# Screening of healthcare workers for SARS-CoV-2 highlights the role of asymptomatic carriage in COVID-19 transmission

Lucy Rivett[1,2†], Sushmita Sridhar[3,4,5†], Dominic Sparkes[1,2†], Matthew Routledge[1,2†], Nick K Jones[1,2,4,5†], Sally Forrest[4,5], Jamie Young[6], Joana Pereira-Dias[4,5], William L Hamilton[1,2], Mark Ferris[7], M Estee Torok[5,8], Luke Meredith[9], The CITIID-NIHR COVID-19 BioResource Collaboration, Martin D Curran[2], Stewart Fuller[10], Afzal Chaudhry[11], Ashley Shaw[10], Richard J Samworth[12], John R Bradley[4,13], Gordon Dougan[4,5], Kenneth GC Smith[4,5], Paul J Lehner[1,4,5], Nicholas J Matheson[1,4,5,14], Giles Wright[7], Ian G Goodfellow[9‡], Stephen Baker[4,5‡], Michael P Weekes[1,4,5‡]*

[1]Department of Infectious Diseases, Cambridge University NHS Hospitals Foundation Trust, Cambridge, United Kingdom; [2]Clinical Microbiology and Public Health Laboratory, Public Health England, Cambridge, United Kingdom; [3]Wellcome Sanger Institute, Hinxton, United Kingdom; [4]Department of Medicine, University of Cambridge, Cambridge, United Kingdom; [5]Cambridge Institute of Therapeutic Immunology and Infectious Disease (CITIID), Jeffrey Cheah Biomedical Centre, Cambridge Biomedical Campus, University of Cambridge, Cambridge, United Kingdom; [6]Academic Department of Medical Genetics, University of Cambridge, Cambridge, United Kingdom; [7]Occupational Health and Wellbeing, Cambridge University Hospitals NHS Foundation Trust, Cambridge, United Kingdom; [8]Department of Microbiology, Cambridge University NHS Hospitals Foundation Trust, Cambridge, United Kingdom; [9]Division of Virology, Department of Pathology, University of Cambridge, Cambridge, United Kingdom; [10]National Institutes for Health Research Cambridge, Clinical Research Facility, Cambridge, United Kingdom; [11]Cambridge University Hospitals NHS Foundation Trust, Cambridge, United Kingdom; [12]Statistical Laboratory, Centre for Mathematical Sciences, Cambridge, United Kingdom; [13]National Institutes for Health Research Cambridge Biomedical Research Centre, Cambridge, United Kingdom; [14]NHS Blood and Transplant, Cambridge, United Kingdom

*For correspondence:
mpw1001@cam.ac.uk

†These authors contributed equally to this work
‡These authors also contributed equally to this work

Group author details:
The CITIID-NIHR COVID-19 BioResource Collaboration See page 17

**Abstract** Significant differences exist in the availability of healthcare worker (HCW) SARS-CoV-2 testing between countries, and existing programmes focus on screening symptomatic rather than asymptomatic staff. Over a 3 week period (April 2020), 1032 asymptomatic HCWs were screened for SARS-CoV-2 in a large UK teaching hospital. Symptomatic staff and symptomatic household contacts were additionally tested. Real-time RT-PCR was used to detect viral RNA from a throat +nose self-swab. 3% of HCWs in the *asymptomatic screening group* tested positive for SARS-CoV-2. 17/30 (57%) were truly asymptomatic/pauci-symptomatic. 12/30 (40%) had experienced symptoms compatible with coronavirus disease 2019 (COVID-19)>7 days prior to testing, most self-isolating, returning well. Clusters of HCW infection were discovered on two independent wards. Viral genome sequencing showed that the majority of HCWs had the dominant lineage B•1. Our

data demonstrates the utility of comprehensive screening of HCWs with minimal or no symptoms. This approach will be critical for protecting patients and hospital staff.

## Introduction

Despite the World Health Organisation (WHO) advocating widespread testing for SARS-CoV-2, national capacities for implementation have diverged considerably (*WHO, 2020b*; *Our World in Data, 2020*). In the UK, the strategy has been to perform SARS-CoV-2 testing for essential workers who are symptomatic themselves or have symptomatic household contacts. This approach has been exemplified by recent studies of symptomatic HCWs (*Hunter et al., 2020*; *Keeley et al., 2020*). The role of nosocomial transmission of SARS-CoV-2 is becoming increasingly recognised, accounting for 12–29% of cases in some reports (*Wang et al., 2020*). Importantly, data suggest that the severity and mortality risk of nosocomial transmission may be greater than for community-acquired COVID-19 (*McMichael et al., 2020*).

Protection of HCWs and their families from the acquisition of COVID-19 in hospitals is paramount, and underscored by rising numbers of HCW deaths nationally and internationally (*Cook et al., 2020*; *CDC COVID-19 Response Team, 2020*). In previous epidemics, HCW screening programmes have boosted morale, decreased absenteeism and potentially reduced long-term psychological sequelae (*McAlonan et al., 2007*). Screening also allows earlier return to work when individuals or their family members test negative (*Hunter et al., 2020*; *Keeley et al., 2020*). Another major consideration is the protection of vulnerable patients from a potentially infectious workforce (*McMichael et al., 2020*), particularly as social distancing is not possible whilst caring for patients. Early identification and isolation of infectious HCWs may help prevent onward transmission to patients and colleagues, and targeted infection prevention and control measures may reduce the risk of healthcare-associated outbreaks.

The clinical presentation of COVID-19 can include minimal or no symptoms (*WHO, 2020a*). Asymptomatic or pre-symptomatic transmission is clearly reported and is estimated to account for around half of all cases of COVID-19 (*He et al., 2020*). Screening approaches focussed solely on symptomatic HCWs are therefore unlikely to be adequate for suppression of nosocomial spread. Preliminary data suggests that mass screening and isolation of asymptomatic individuals can be an effective method for halting transmission in community-based settings (*Day, 2020*). Recent modelling has suggested that weekly testing of asymptomatic HCWs could reduce onward transmission by 16–23%, on top of isolation based on symptoms, provided results are available within 24 hr (*Imperial College COVID-19 Response Team, 2020*). The need for widespread adoption of an expanded screening programme for asymptomatic, as well as symptomatic HCWs, is apparent (*Imperial College COVID-19 Response Team, 2020*; *Black et al., 2020*; *Gandhi et al., 2020*).

Challenges to the roll-out of an expanded screening programme include the ability to increase diagnostic testing capacity, logistical issues affecting sampling and turnaround times and concerns about workforce depletion should substantial numbers of staff test positive. Here, we describe how we have dealt with these challenges and present initial findings from a comprehensive staff screening programme at Cambridge University Hospitals NHS Foundation Trust (CUHNFT). This has included systematic screening of >1000 asymptomatic HCWs in their workplace, in addition to >200 symptomatic staff or household contacts. Screening was performed using a validated real-time reverse transcription PCR (RT-PCR) assay detecting SARS-CoV-2 from combined oropharyngeal (OP) and nasopharyngeal (NP) swabs (*Sridhar et al., 2020*). Rapid viral sequencing of positive samples was used to further assess potential epidemiological linkage where nosocomial transmission was suspected. Our experience highlights the value of programmes targeting both symptomatic and asymptomatic staff, and will be informative for the establishment of similar programmes in the UK and globally.

## Results

### Characteristics of HCW and testing groups

Between 6[th] and 24[th] April 2020, 1,270 HCWs in CUHNFT and their symptomatic household contacts were swabbed and tested for SARS-CoV-2 by real-time RT-PCR. The median age of the HCWs was

**eLife digest** Patients admitted to NHS hospitals are now routinely screened for SARS-CoV-2 (the virus that causes COVID-19), and isolated from other patients if necessary. Yet healthcare workers, including frontline patient-facing staff such as doctors, nurses and physiotherapists, are only tested and excluded from work if they develop symptoms of the illness.

However, there is emerging evidence that many people infected with SARS-CoV-2 never develop significant symptoms: these people will therefore be missed by 'symptomatic-only' testing. There is also important data showing that around half of all transmissions of SARS-CoV-2 happen before the infected individual even develops symptoms. This means that much broader testing programs are required to spot people when they are most infectious.

Rivett, Sridhar, Sparkes, Routledge et al. set out to determine what proportion of healthcare workers was infected with SARS-CoV-2 while also feeling generally healthy at the time of testing. Over 1,000 staff members at a large UK hospital who felt they were well enough to work, and did not fit the government criteria for COVID-19 infection, were tested. Amongst these, 3% were positive for SARS-CoV-2. On closer questioning, around one in five reported no symptoms, two in five very mild symptoms that they had dismissed as inconsequential, and a further two in five reported COVID-19 symptoms that had stopped more than a week previously. In parallel, healthcare workers with symptoms of COVID-19 (and their household contacts) who were self-isolating were also tested, in order to allow those without the virus to quickly return to work and bolster a stretched workforce.

Finally, the rates of infection were examined to probe how the virus could have spread through the hospital and among staff – and in particular, to understand whether rates of infection were greater among staff working in areas devoted to COVID-19 patients. Despite wearing appropriate personal protective equipment, healthcare workers in these areas were almost three times more likely to test positive than those working in areas without COVID-19 patients. However, it is not clear whether this genuinely reflects greater rates of patients passing the infection to staff. Staff may give the virus to each other, or even acquire it at home.

Overall, this work implies that hospitals need to be vigilant and introduce broad screening programmes across their workforces. It will be vital to establish such approaches before 'lockdown' is fully lifted, so healthcare institutions are prepared for any second peak of infections.

---

34; 71% were female and 29% male. The technical RT-PCR failure rate was 2/1,270 (0•2% see Materials and methods); these were excluded from the 'Tested' population for further analysis. Ultimately, 5% (n = 61) of swabs were SARS-CoV-2 positive. 21 individuals underwent repeat testing for a variety of reasons, including evolving symptoms (n = 3) and scoring 'medium' probability on clinical COVID-19 criteria (*Tables 1–2*) (n = 11). All remained SARS-CoV-2 negative. Turn around time from sample collection to resulting was 12–36 hr; this varied according to the time samples were obtained.

*Table 3* outlines the total number of SARS-CoV-2 tests performed in each screening group (*HCW asymptomatic*, *HCW symptomatic*, and *HCW symptomatic household contact*) categorised according to the ward with the highest anticipated risk of exposure to COVID-19 ('red', high; 'amber', medium; 'green', low; *Tables 4–5*). In total, 31/1,032 (3%) of those tested in the *HCW asymptomatic screening group* tested SARS-CoV-2 positive. In comparison, 30/221 (14%) tested positive when *HCW symptomatic* and *HCW symptomatic household contact screening groups* were combined. As expected, symptomatic HCWs and their household contacts were significantly more likely to test positive than HCWs from the *asymptomatic screening group* (p<0•0001, Fisher's exact test). HCWs working in 'red' or 'amber' wards were significantly more likely to test positive than those working in 'green' wards (p=0•0042, Fisher's exact test).

All users of FFP3 masks underwent routine fit-testing prior to usage. Cleaning and re-use of masks, theatre caps, gloves, aprons or gowns was actively discouraged. Cleaning and re-use of eye protection was permitted for certain types of goggles and visors, as specified in the hospital's PPE protocol. Single-use eye protection was in use in most Scenario 1 and 2 areas, and was not cleaned and re-used. All non-invasive ventilation or use of high-flow nasal oxygen on laboratory-confirmed or

clinically suspected COVID-19 patients was performed in negative-pressure (−5 pascals) side rooms, with 10 air changes per hour and use of Scenario 2 PPE. All other aerosol generating procedures were undertaken with Scenario 2 PPE precautions, in negative- or neutral- pressure facilities. General clinical areas underwent a minimum of 6 air changes per hour, but all critical care areas underwent a minimum of 10 air changes per hour as a matter of routine. Surgical operating theatres routinely underwent a minimum of 25 air changes per hour.

Viral loads varied between individuals, potentially reflecting the nature of the sampling site. However, for individuals testing positive for SARS-CoV-2, viral loads were significantly lower for those in the *HCW asymptomatic screening group* than in those tested due to the presence of symptoms (*Figure 1*). For the *HCW symptomatic* and *HCW symptomatic contact screening groups*, viral loads did not correlate with duration of symptoms or with clinical criteria risk score (*Figure 1—figure supplement 1* and data not shown).

## Three subgroups of SARS-CoV-2 positive asymptomatic HCW

Each individual in the *HCW asymptomatic screening group* was contacted by telephone to establish a clinical history, and COVID-19 probability criteria (*Table 1*) were retrospectively applied to categorise any symptoms in the month prior to testing (*Figure 2*). One HCW could not be contacted to obtain further history. Individuals captured by the *HCW asymptomatic screening group* were generally *asymptomatic at the time of screening*, however could be divided into three sub-groups: (i) HCWs with no symptoms at all, (ii) HCWs with (chiefly low-to-medium COVID-19 probability) symptoms commencing ≤7 days prior to screening and (iii) HCWs with (typically high COVID-19 probability) symptoms commencing >7 days prior to screening (*Figure 2*). 9/12 (75%) individuals with symptom onset >7 days previously had appropriately self-isolated and then returned to work. One individual with no symptoms at the time of swabbing subsequently developed symptoms prior to being contacted with their positive result. Overall, 5/1032 (0.5%) individuals in the asymptomatic screening group were identified as truly asymptomatic carriers of SARS-CoV-2, and 1/1032 (0.1%) was identified as pre-symptomatic. *Box 1* shows illustrative clinical vignettes.

## Identification of two clusters of cases by screening asymptomatic HCWs

For the *HCW asymptomatic screening group*, nineteen wards were identified for systematic priority screening as part of hospital-wide surveillance. Two further areas were specifically targeted for screening due to unusually high staff sickness rates (ward F), or concerns about appropriate PPE

**Table 1.** Clinical criteria for estimating pre-test probability of COVID-19.

**COVID-19 probability criteria**

| | |
|---|---|
| Major | Fever (>37.8 ˚C) |
| | New persistent cough |
| | Unprotected close contact with a confirmed case* |
| Minor | Hoarse voice |
| | Non-persistent cough |
| | Sore throat |
| | Nasal discharge or congestion |
| | Shortness of breath |
| | Wheeze |
| | Headache |
| | Muscle aches |
| | Nausea and/or vomiting and/or diarrhoea |
| | Loss of sense of taste or smell |

*Unprotected close contact defined as either face-to-face contact or spending more than 15 min within 2 metres of an infected person, without wearing appropriate personal protective equipment (PPE).

Table 2. Categories of pre-test probability of COVID-19, according to the presence of clinical features shown in **Table 1**.

| Stratification of COVID-19 probability | | Implications for exclusion from work |
|---|---|---|
| High probability | ≥2 major symptoms or ≥1 major symptom and ≥ 2 minor symptoms | Self-isolate for 7 days from the date of onset, regardless of the test result. Only return to work if afebrile for 48 hr and symptoms have improved*.<br>Household contacts should self-quarantine for 14 days from the date of symptom onset in the index case, regardless of the test result. If they develop symptoms, they should self-isolate for 7 days from the date of onset, and only return to work if afebrile for 48 hr and symptoms have improved*. |
| Medium probability | 1 major symptom or 0 major symptoms and ≥ 3 minor symptoms | Test result positive: self-isolate for 7 days from the date of onset, and only return to work if afebrile for 48 hr and symptoms have improved*. Household contacts should self-quarantine for 14 days from the date of index case symptom onset. If they develop symptoms, they should self-isolate for 7 days from the date of onset, and only return to work if afebrile for 48 hr and symptoms have improved*.<br>Test result negative: repeat testing at 48 hr from the initial swab. If repeat testing is positive, follow the advice detailed above. If repeat testing is negative, return to work, unless symptoms worsen. Self-quarantine not required for household contacts. |
| Low probability | 0 major symptoms and 1–2 minor symptoms | Test result positive: self-isolate for 7 days from the date of test, and only return to work if afebrile for 48 hr and symptoms have improved*. Household contacts should self-quarantine for 14 days from the date of test. If they develop symptoms, they should self-isolate for 7 days from the date of onset, and only return to work if afebrile for 48 hr and symptoms have improved*.<br>Test result negative: return to work, unless symptoms worsen. Self-quarantine not required for household contacts. |
| Asymptomatic | 0 major symptoms and 0 minor symptoms | Test result positive: self-isolate for 7 days from the date of test. If symptoms develop after the test, self-isolation should occur for 7 days from the date of onset, and return to work should only occur if afebrile for 48 hr and symptoms have improved*. Household contacts should self-quarantine for 14 days from the date of the test. If they develop symptoms, they should self-isolate for 7 days from the date of onset, and only return to work if afebrile for 48 hr and symptoms have improved*.<br>Test result negative: continue working, unless symptoms develop. Self-quarantine not required for household contacts. |

*Residual cough in the absence of other symptoms should not preclude returning to work.

usage (ward Q) (**Figure 3**). Interestingly, in line with findings in the total HCW population, a significantly greater proportion of HCWs working on 'red' wards compared to HCWs working on 'green' wards tested positive as part of the asymptomatic screening programme ('green' 6/310 vs 'red' 19/372; p=0.0389, Fisher's exact test). The proportion of HCW with a positive test was significantly higher on Ward F than on other wards categorised as 'green' clinical areas (ward F 4/43 vs other 'green' wards 2/267; p=0.0040, Fisher's exact test). Likewise, amongst wards in the 'red' areas, ward Q showed significantly higher rates of positive HCW test results (ward Q 7/37 vs other 'red' wards 12/335; p=0.0011, Fisher's exact test).

Ward F is an elderly care ward, designated as a 'green' area with Scenario 0 PPE (**Tables 4–5**), with a high proportion of COVID-19 vulnerable patients due to age and comorbidity. 4/43 (9%) ward staff tested positive for SARS-CoV-2. In addition, two staff members on this ward tested positive in the *HCW symptomatic/symptomatic contact screening groups*. All positive HCWs were requested to self-isolate, the ward was closed to admissions and escalated to Scenario 1 PPE (**Table 5**). Reactive screening of a further 18 ward F staff identified an additional three positive asymptomatic HCWs (**Figure 4**). Sequence analysis indicated that 6/9 samples from HCW who worked on ward F belonged to SARS-CoV-2 lineage B.1 (currently known to be circulating in at least 43 countries [**Rambaut et al., 2020**]), with a further two that belonged to B1.7 and one that belonged to B2.1. This suggests more than two introductions of SARS-CoV-2 into the HCW population on ward F (**Figure 4—figure supplements 1–2, Table 6**). It was subsequently found that two

**Table 3.** Total number of SARS-CoV-2 tests performed in each screening group categorised according to the highest risk ward of potential exposure.

| | Clinical area | | | | |
|---|---|---|---|---|---|
| | Green | Amber | Red | Unknown | Total |
| HCW asymptomatic screening group | 7/454 (1.5%) | 4/78 (5.1%) | 20/466 (4.3%) | 0/34 (0%) | 31/1032 (3%) |
| HCW symptomatic screening group | 8/66 (12.1%) | 1/9 (11.1%) | 17/88 (19.3%) | 0/6 (0%) | 26/169 (15.4%) |
| HCW symptomatic household contacts | 2/14 (14.3%) | 0/1 (0%) | 0/14 (0%) | 2/23 (8.7%) | 4/52 (7.7%) |
| Unknown | 0/4 (0%) | 0/0 | 0/7 (0%) | 0/4 (0%) | 0/15 (0%) |
| All | 17/538 (3.2%) | 5/88 (5.7%) | 37/575 (6.4%) | 2/67 (3%) | 61/1268 (4.8%) |

further staff members from ward F had previously been admitted to hospital with severe COVID-19 infection.

Ward Q is a general medical ward designated as a 'red' clinical area for the care of COVID-19 positive patients, with a Scenario 1 PPE protocol (*Tables 4–5*). Here, 7/37 (19%) ward staff tested positive for SARS-CoV-2. In addition, one staff member tested positive as part of the *HCW symptomatic screening group*, within the same period as ward surveillance. Reactive screening of a further five staff working on Ward Q uncovered one additional infection. 4/4 sequenced viruses were of the B.1 lineage (*Figure 4—figure supplements 1–2*, *Table 6*; other isolates could not be sequenced due to a sample CT value >30). All positive HCWs were requested to self-isolate, and infection control and PPE reviews were undertaken to ensure that environmental cleaning and PPE donning/doffing practices were compliant with hospital protocol. Staff training and education was provided to address observed instances of incorrect infection control or PPE practice.

Ward O, a 'red' medical ward, had similar numbers of asymptomatic HCWs screened as ward F, and a similar positivity rate (4/44; 9%). This ward was listed for further cluster investigation after the study ended, however incorrect PPE usage was not noted during the study period.

## Characteristics of the HCW symptomatic and HCW symptomatic-contact screening groups

The majority of individuals who tested positive for SARS-CoV-2 after screening due to the presence of symptoms had high COVID-19 probability (*Table 7*). This reflects national guidance regarding self-isolation at the time of our study (*UK Government, 2020a*).

## Discussion

Through the rapid establishment of an expanded HCW SARS-CoV-2 screening programme, we discovered that 31/1,032 (3%) of HCWs tested positive for SARS-CoV-2 in the absence of symptoms. Of 30 individuals from this *asymptomatic screening group* studied in more depth, 6/30 (20%) had not experienced any symptoms at the time of their test. 1/6 became symptomatic suggesting that the true asymptomatic carriage rate was 5/1,032 (0.5%). 11/30 (37%) had experienced mild symptoms prior to testing. Whilst temporally associated, it cannot be assumed that these symptoms necessarily resulted from COVID-19. These proportions are difficult to contextualise due to paucity of

**Table 4.** The hospital's traffic-light colouring system for categorising wards according to anticipated COVID-19 exposure risk. Different types of PPE were used in each (*Table 5*).

| Red (high risk) | Amber (medium risk) | Green (low risk) |
|---|---|---|
| Areas with confirmed SARS-CoV-2 RT-PCR positive patients, or patients with very high clinical suspicion of COVID-19 | Areas with patients awaiting SARS-CoV-2 RT-PCR test results, or that have been exposed and may be incubating infection | Areas with no known SARS-CoV-2 RT-PCR positive patients, and none with clinically suspected COVID-19 |

**Table 5.** PPE protocols ('Scenarios') and their relation to ward category.

| | PPE 'Scenarios' | | | |
| --- | --- | --- | --- | --- |
| | Scenario 0 | Scenario 1 | Scenario 2 | Scenario 3 |
| Area description | All clinical areas without any known or suspected COVID-19 cases | Designated ward, triage and assessment-based care with suspected or confirmed COVID-19 patients | Cohorted areas where aerosol-generating procedures are carried out frequently with suspected or confirmed COVID-19 patients | Operating theatres where procedures are performed with suspected or confirmed COVID-19 patients |
| PPE description | Fluid resistant face mask at all times, apron and non-sterile gloves for patient contact (within two metres) | Surgical scrubs, fluid resistant face mask, theatre cap, eye protection, apron and non-sterile gloves | Water repellent gown, FFP3 mask, eye protection, theatre cap, surgical gloves, with an apron and non-sterile gloves in addition for patient contact (within two metres) | Water repellent gown, FFP3 mask, eye protection, theatre cap and surgical gloves |
| Ward categories | Green wards, for example designated areas of emergency department and medical admissions unit. Medical, surgical and haematology wards/outpatient clinics. | Amber + red wards, for example designated areas of emergency department and medical admissions unit. Designated CoVID-19 confirmed wards. | Amber + red wards, for example intensive care unit, respiratory units with non-invasive ventilation facilities. | All operating theatres, including facilities for bronchoscopy and endoscopy. |

point-prevalence data from asymptomatic individuals in similar healthcare settings or the wider community. For contrast, 60% of asymptomatic residents in a recent study tested positive in the midst of a care home outbreak (*Arons et al., 2020*). Regardless of the proportion, however, many secondary and tertiary hospital-acquired infections were undoubtedly prevented by identifying and isolating these SARS-CoV-2 positive HCWs.

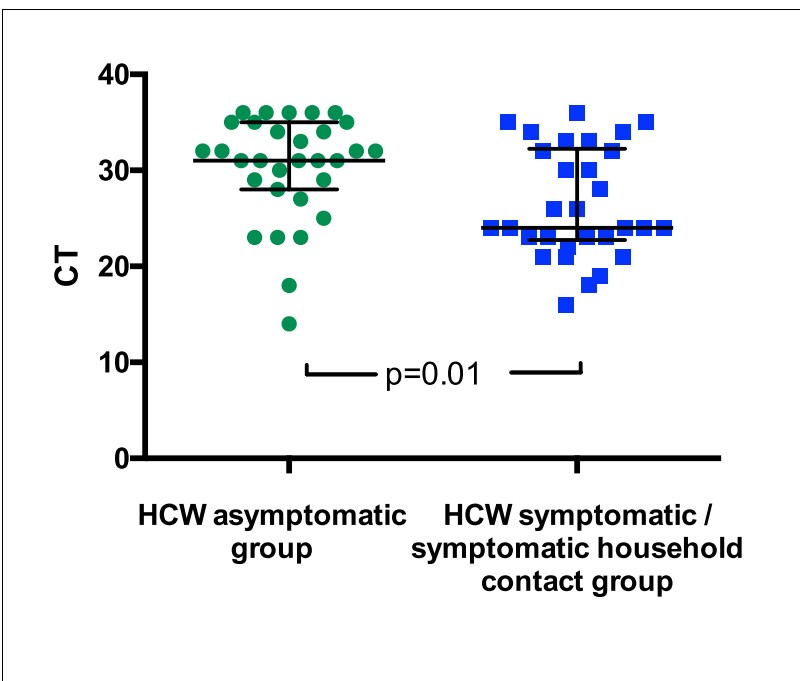

**Figure 1.** SARS-CoV-2 RNA CT (cycle threshold) values for those individuals who tested positive shown according to HCW group. *HCW asymptomatic screening group* (green circles); *HCW symptomatic* or *symptomatic household contact screening groups* (blue squares). A Mann Whitney test was used to compare the two groups. Bars: median + / - interquartile range. Note that lower CT values correspond to earlier detection of the viral RNA in the RT-PCR process and therefore identify swabs with higher numbers of copies of the viral genome.
The online version of this article includes the following figure supplement(s) for figure 1:

**Figure supplement 1.** SARS-CoV-2 RNA CT values for HCWs testing positive according to presence and duration of symptoms.

## Box 1. Clinical vignettes.

Self-isolation instructions were as described in *Table 2*.

**Case 1**: **Completely asymptomatic**. HCW1 had recently worked on four wards (two 'green', two 'amber'). Upon testing positive, she reported no symptoms over the preceding three weeks, and was requested to go home and self-isolate immediately. HCW1 lived with her partner who had no suggestive symptoms. Upon follow-up telephone consultation 14 days after the test, HCW1 had not developed any significant symptoms, suggesting true asymptomatic infection.

**Case 2**: **Pre-symptomatic**. HCW2 was swabbed whilst asymptomatic, testing positive. When telephoned with the result, she reported a cough, fever and headache starting within the last 24 hr and was advised to self-isolate from the time of onset of symptoms (*Table 2*). Her partner, also a HCW, was symptomatic and had been confirmed as SARS-CoV-2 positive 2 days previously, suggesting likely transmission of infection to HCW2.

**Case 3**: **Low clinical probability of COVID** HCW3 developed mild self-limiting pharyngitis three days prior to screening and continued to work in the absence of cough or fever. She had been working in' green' areas of the hospital, due to a background history of asthma. Self-isolation commenced from the time of the positive test. HCW3's only contact outside the hospital, her housemate, was well. On follow-up telephone consultation, HCW3's mild symptoms had fully resolved, with no development of fever or persistent cough, suggesting pauci-symptomatic infection.

**Case 4**: **Medium clinical probability of COVID** HCW4 experienced anosmia, nausea and headache three days prior to screening, and continued to work in the absence of cough or fever. Self-isolation commenced from the time of the positive test. One son had experienced a mild cough ~3 weeks prior to HCW4's test, however her partner and other son were completely asymptomatic. Upon follow-up telephone consultation 10 days after the test, HCW4's mild symptoms had not progressed, but had not yet resolved.

**Case 5**: **High clinical probability of COVID**. HCW5 had previously self-isolated, and did not repeat this in the presence of new high-probability symptoms six days before screening. Self-isolation commenced from the date of the new symptoms with the caveat that they should be completely well for 48 hr prior to return to work. All household contacts were well. However, another close colleague working on the same ward had also tested positive, suggesting potential transmission between HCWs on that ward.

12/30 (40%) individuals from the *HCW asymptomatic screening group* reported symptoms > 7 days prior to testing, and the majority experiencing symptoms consistent with a high probability of COVID-19 had appropriately self-isolated during that period. Patients with COVID-19 can remain SARS-CoV-2 PCR positive for a median of 20 days (IQR 17–24) after symptom onset (*Zhou et al., 2020*), and the limited data available suggest viable virus is not shed beyond eight days (*Wölfel et al., 2020*). A pragmatic approach was taken to allowing individuals to remain at work, where the HCW had experienced high probability symptoms starting >7 days and ≤1 month prior to their test and had been well for the preceding 48 hr. This approach was based on the following: low seasonal incidence of alternative viral causes of high COVID-19 probability symptoms in the UK (*Public Health England, 2018*), the high potential for SARS-CoV-2 exposure during the pandemic and the potential for prolonged, non-infectious shedding of viral RNA (*Zhou et al., 2020*; *Wölfel et al., 2020*). For other individuals, we applied standard national guidelines requiring isolation for seven days from the point of testing (*UK Government, 2020b*). However, for HCW developing symptoms after a positive swab, isolation was extended for seven days from symptom onset.

Our data clearly demonstrate that focusing solely on the testing of individuals fitting a strict clinical case definition for COVID-19 will inevitably miss asymptomatic and pauci-symptomatic disease. This is of particular importance in the presence of falling numbers of community COVID-19 cases, as hospitals will become potential epicentres of local outbreaks. Therefore, we suggest that in the setting of limited testing capacity, a high priority should be given to a reactive asymptomatic screening programme that responds in real-time to HCW sickness trends, or (to add precision) incidence of positive tests by area. The value of this approach is illustrated by our detection of a cluster of cases in ward F, where the potential for uncontrolled staff-to-staff or staff-to-patient transmission could have led to substantial morbidity and mortality in a particularly vulnerable patient group. As SARS-CoV-2 testing capacity increases, rolling programmes of serial screening for asymptomatic staff in all

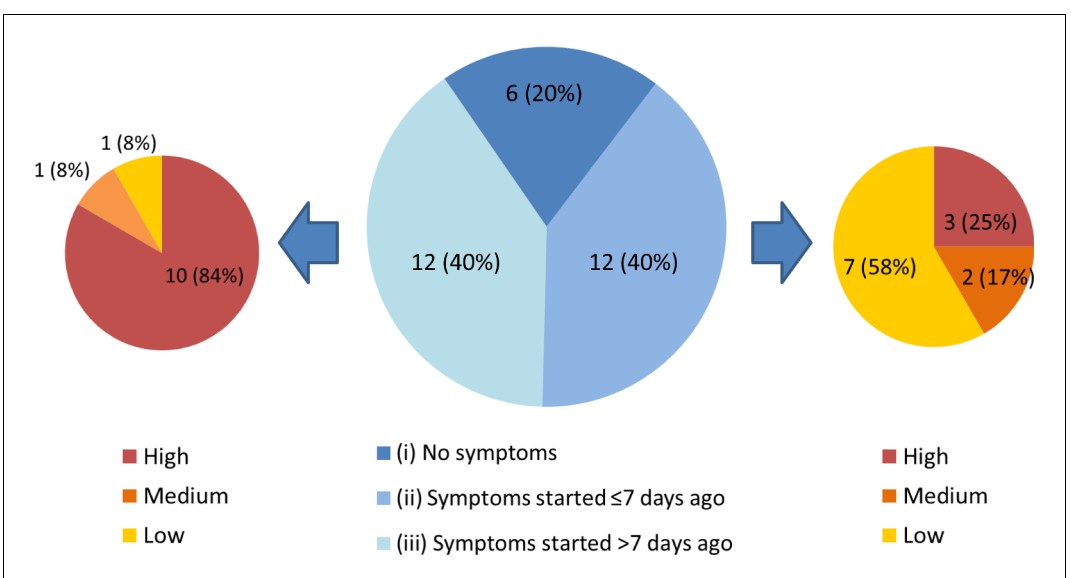

**Figure 2.** Three subgroups of staff testing SARS-CoV-2 positive from the *HCW asymptomatic screening group.* (central pie chart, described in detail in the main text). n = number of individuals (% percentage of total). The peripheral pie charts show number and percentage of individuals in groups (ii – right pie chart) and (iii – left pie chart) with low, medium and high COVID-19 probability symptoms upon retrospective analysis.

areas of the hospital is recommended, with the frequency of screening being dictated by anticipated probability of infection. The utility of this approach in care-homes and other essential institutions should also be explored, as should serial screening of long-term inpatients.

The early success of our programme relied upon substantial collaborative efforts between a diverse range of local stakeholders. Similar collaborations will likely play a key role in the rapid, de novo development of comprehensive screening programmes elsewhere. The full benefits of enhanced HCW screening are critically dependent upon rapid availability of results. A key success of our programme has been bespoke optimisation of sampling and laboratory workflows enabling same-day resulting, whilst minimising disruption to hospital processes by avoiding travel to off-site testing facilities. Rapid turnaround for testing and sequencing is vital in enabling timely response to localised infection clusters, as is the maintenance of reserve capacity to allow urgent, reactive investigations.

There appeared to be a significantly higher incidence of HCW infections in 'red' compared to 'green' wards. Many explanations for this observation exist, and this study cannot differentiate between them. Possible explanations include transmission between patients and HCW, HCW-to-HCW transmission, variability of staff exposure outside the workplace and non-random selection of wards. It is also possible that, even over the three weeks of the study, 'red' wards were sampled earlier during the evolution of the epidemic when transmission was greater. Further research into these findings is clearly needed on a larger scale. Furthermore, given the clear potential for pre-symptomatic and asymptomatic transmission amongst HCWs, and data suggesting that infectivity may peak prior to symptom onset (*He et al., 2020*), there is a strong argument for basic PPE provision in all clinical areas.

The identification of transmission within the hospital through routine data is problematic. Hospitals are not closed systems and are subject to numerous external sources of infection. Coronaviruses generally have very low mutation rates ($\sim 10^{-6}$ per site per cycle) (*Sanjuán et al., 2010*), with the first reported sequence of the current pandemic only published on 12[th] January 2020 (*GenBank, 2020*). In addition, given SARS CoV-2 was only introduced into the human population in late 2019, there is at present a lack of diversity in circulating strains. However, as the pandemic unfolds and detailed epidemiological and genome sequence data from patient and HCW clusters are generated, real-time study of transmission dynamics will become an increasingly important means of informing disease control responses and rapidly confirming (or refuting) hospital acquired infection. Importantly,

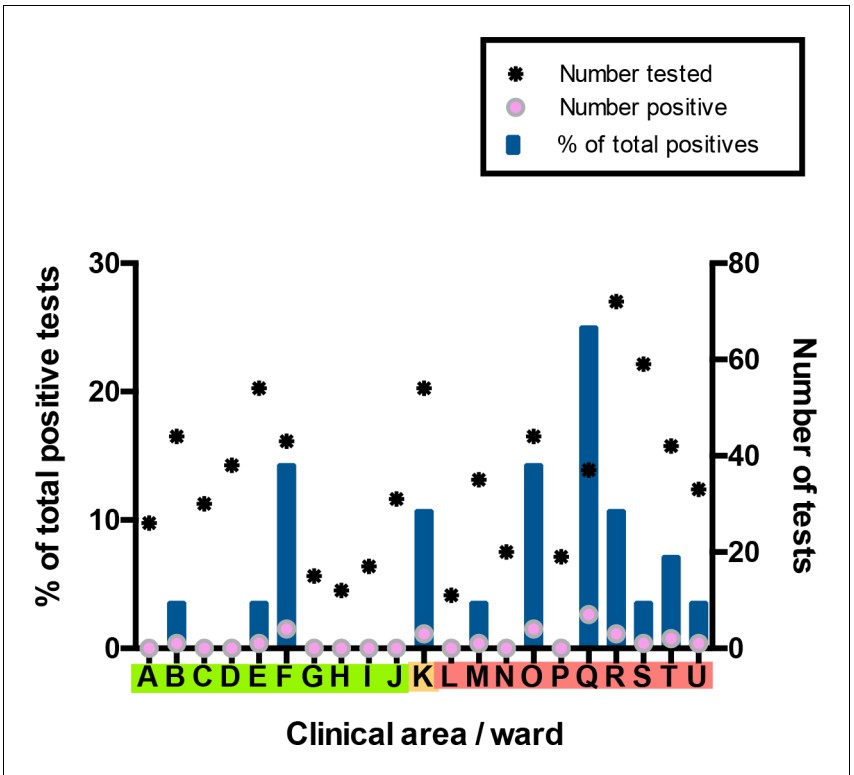

**Figure 3.** Distribution of SARS-CoV-2 positive cases across 21 clinical areas, detected by ward-based asymptomatic screening. (underlying data shown in 'Source Data'). Wards are coloured ('green', 'amber', 'red') according to risk of anticipated exposure to COVID-19 (*Table 4*). HCWs working across >1 ward were counted for each area. The left-hand y-axis shows the percentage of positive results from a given ward compared to the total positive results from the *HCW asymptomatic screening group* (blue bars). The right-hand y-axis shows the total number of SARS-CoV-2 tests (stars) and the number positive (pink circles). Additional asymptomatic screening tests were subsequently performed in an intensified manner on ward F and ward Q after identification of clusters of positive cases on these wards (*Figure 4*). Asymptomatic screening tests were also performed for a number of individuals from other clinical areas on an opportunistic basis; none of these individuals tested positive. Results of these additional tests are included in summary totals in *Table 1*, but not in this figure.

implementation of such a programme would require active screening and rapid sequencing of positive cases in both the HCW and patient populations. Prospective epidemiological data will also inform whether hospital staff are more likely to be infected in the community or at work, and may identify risk factors for the acquisition of infection, such as congregation in communal staff areas or inadequate access to PPE.

Our study is limited by the relatively short time-frame, a small number of positive tests and a lack of behavioural data. In particular, the absence of detailed workplace and community epidemiological data makes it difficult to draw firm conclusions with regards to hospital transmission dynamics. The low rate of observed positive tests may be partly explained by low rates of infection in the East of England in comparison with other areas of the UK (cumulative incidence 0.17%, thus far) (*Public Health England, 2020*). The long-term benefits of HCW screening on healthcare systems will be informed by sustained longitudinal sampling of staff in multiple locations. More comprehensive data will parametrise workforce depletion and COVID-19 transmission models. The incorporation of additional information including staffing levels, absenteeism, and changes in proportions of staff self-isolating before and after the introduction of widespread testing will better inform the impact of screening at a national and international level. Such models will be critical for optimising the impact on occupationally-acquired COVID-19, and reducing the likelihood that hospitals become hubs for sustained COVID-19 transmission.

In the absence of an efficacious vaccine, additional waves of COVID-19 are likely as social distancing rules are relaxed. Understanding how to limit hospital transmission will be vital in determining

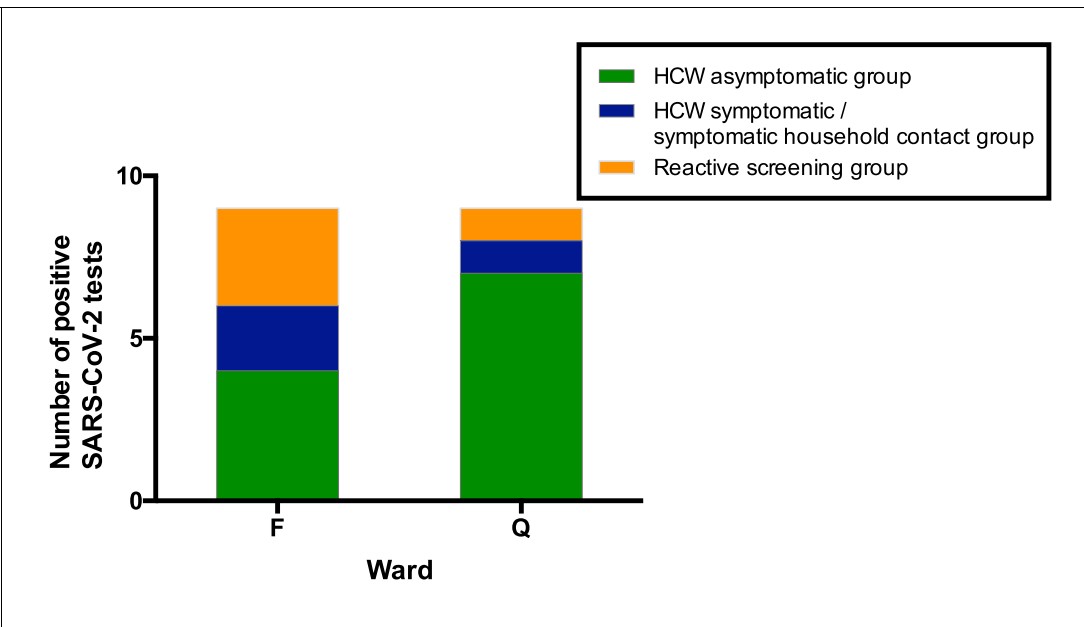

**Figure 4.** All SARS-CoV-2 positive HCW identified in Wards F and Q, stratified by their mechanism of identification. Individuals testing positive for SARS-CoV-2 in the '*HCW asymptomatic screening group*' were identified by the asymptomatic screening programme. Individuals testing positive in the '*HCW symptomatic/symptomatic household contact groups*' were identified by self-presentation after developing symptoms. Individuals testing positive in the '*Reactive screening group*' were identified by an intensified screening programme after initial positive clusters had been recognised. The online version of this article includes the following figure supplement(s) for figure 4:

**Figure supplement 1.** Further details of sequencing data.
**Figure supplement 2.** Phylogenetic tree of 34 healthcare worker (HCW) SARS-CoV-2 genomes.

infection control policy, and retain its relevance when reliable serological testing becomes widely available. Our data suggest that the roll-out of screening programmes to include asymptomatic as well as symptomatic patient-facing staff should be a national and international priority. Our approach may also be of benefit in reducing transmission in other institutions, for example care-homes. Taken together, these measures will increase patient confidence and willingness to access healthcare services, benefiting both those with COVID-19 and non-COVID-19 disease.

## Materials and methods

### Staff screening protocols

Two parallel streams of entry into the testing programme were established and managed jointly by the Occupational Health and Infectious Diseases departments. The first (*HCW symptomatic,* and *HCW symptomatic household contact screening groups*) allowed any patient-facing or non-patient-facing hospital employee (HCW) to refer themselves or a household contact, including children, should they develop symptoms suggestive of COVID-19. The second (*HCW asymptomatic screening group*) was a rolling programme of testing for all patient-facing and non-patient-facing staff working in defined clinical areas thought to be at risk of SARS-CoV-2 transmission. Daily workforce sickness reports and trends in the results of HCW testing were monitored to enable areas of concern to be highlighted and targeted for screening and cluster analysis, in a reactive approach. High throughput clinical areas where staff might be exposed to large numbers of suspected COVID-19 patients were also prioritised for staff screening. These included the Emergency Department, the COVID-19 Assessment Unit, and a number of 'red' inpatient wards. Staff caring for the highest priory 'shielding' patients (Haematology/Oncology, Transplant medicine) were also screened, as were a representative sample of staff from 'Amber' and 'Green' areas. The personal protective equipment (PPE) worn by staff in these areas is summarised in *Table 5*. Inclusion into the programme was voluntary, and offered to all individuals working in a given ward during the time of sampling. Regardless of the

Table 6. Details of each SARS-CoV-2 positive isolate from all HCWs and household contacts in the study.

| Patient ID | Type | HCW_ward | Ct value | Seq Attempted | Seq_ID | % Sequence Coverage | Average Seq Depth | PANGOLIN lineage |
|---|---|---|---|---|---|---|---|---|
| C1 | Symptomatic Contact | HCW Contact | 23.9 | Y | CAMB-7FBB0 | 99.61 | 2048.5 | B.1 |
| C3 | Symptomatic Contact | HCW Contact | 23 | N | | | | Not available |
| H3 | Asymptomatic | B | 31 | Y | CAMB-7C0C3 | 98.61 | 835.084 | B.1 |
| H54 | Symptomatic | B | 35 | | | | | |
| H12 | Symptomatic | C | 16 | Y | CAMB-7FB92 | 99.60 | 3312.22 | B.1 |
| H19 | Asymptomatic | E | 27 | Y | CAMB-7FC26 | 99.61 | 3632.26 | B.1.1 |
| C2 | Asymptomatic | F | 15.5 | Y | CAMB-7FC08 | 99.60 | 3157.08 | B.1 |
| H17 | Asymptomatic | F | 33.6 | Y | CAMB-7FBFC | 99.61 | 1167.76 | B.1 |
| H20 | Asymptomatic | F | 18 | Y | CAMB-7FC35 | 99.61 | 1350.65 | B.1 |
| H21 | Asymptomatic | F | 22.8 | Y | CAMB-7FC44 | 99.60 | 3584.79 | B.1 |
| H22 | Symptomatic | F | 24 | Y | CAMB-7FC53 | 99.60 | 3692.14 | B.1.7 |
| H23 | Asymptomatic | F | 32.7 | Y | CAMB-7FC62 | 99.60 | 1610.33 | B.2.1 |
| H35 | Symptomatic | F | 36 | Y | CAMB-8221F | 73.00 | 104.391 | B.1 |
| H36 | Asymptomatic | F | 29 | Y | CAMB-8222E | 98.59 | 1882.65 | B.1.7 |
| H53 | Symptomatic Contact | HCW Contact | 23 | | | | | |
| H38 | Asymptomatic | K | 36 | N | | | | Not available |
| H39 | Asymptomatic | K | 31 | N | | | | Not available |
| H28 | Symptomatic | K/R/L/T/OTHER | 18 | Y | CAMB-7FD32 | 99.60 | 3770.36 | B.1.11 |
| H11 | Asymptomatic | M | 32 | Y | CAMB-7FB83 | 99.60 | 1044.43 | B.1 |
| H32 | Symptomatic | N | 33 | Y | CAMB-81007 | 97.62 | 1196.53 | B.1 |
| H47 | Symptomatic | N | 32 | N | | | | Not available |
| H31 | Asymptomatic | O | 29 | Y | CAMB-80FFC | 99.59 | 2286.08 | B.1 |
| H45 | Asymptomatic | O | 36 | N | | | | Not available |
| H51 | Symptomatic | O | 33 | | | | | |
| H57 | Symptomatic Contact | O | 23 | | | | | |
| H1 | Asymptomatic | OTHER | 23 | Y | CAMB-7C0A5 | 98.61 | 2277.92 | B.1 |
| H6 | Symptomatic | OTHER | 30 | Y | CAMB-7FB29 | 98.75 | 1317.43 | B.1 |
| H7 | Symptomatic | OTHER | 26 | Y | CAMB-7FB47 | 99.61 | 3599.59 | B.1 |
| H10 | Symptomatic | OTHER | 22 | Y | CAMB-7FB74 | 99.60 | 187.059 | B.1 |

*Table 6 continued on next page*

*Table 6 continued*

| Patient ID | Type | HCW_ward | Ct value | Seq Attempted | Seq_ID | % Sequence Coverage | Average Seq Depth | PANGOLIN lineage |
|---|---|---|---|---|---|---|---|---|
| H14 | Symptomatic | OTHER | 34 | Y | CAMB-7FBCF | 99.61 | 1066.74 | B.1 |
| H16 | Asymptomatic | OTHER | 27.8 | Y | CAMB-7FBED | 99.60 | 796.874 | B.1 |
| H24 | Symptomatic | OTHER | 21 | Y | CAMB-7FC80 | 98.62 | 916.884 | B.1 |
| H25 | Symptomatic | OTHER | 21 | Y | CAMB-7FC9F | 99.60 | 1505.09 | B.1 |
| H33 | Symptomatic | OTHER | 35 | Y | CAMB-81016 | 90.92 | 233.779 | B.1 |
| H40 | Symptomatic | OTHER | 23 | N | | | | Not available |
| H46 | Asymptomatic | OTHER | 36 | N | | | | Not available |
| H55 | Symptomatic | Other | 26 | | | | | |
| H56 | Asymptomatic | Other | 32 | | | | | |
| H30 | Asymptomatic | OTHER/K/O/F | 31 | Y | CAMB-80FDE | 98.61 | 1773.74 | B.1 |
| H5 | Symptomatic | Q | 24 | Y | CAMB-7C1A2 | 97.75 | 2342.24 | B.1 |
| H8 | Symptomatic | Q | 14 | Y | CAMB-7FB56 | 99.60 | 2452.25 | B.1 |
| H18 | Asymptomatic | Q | 30 | Y | CAMB-7FC17 | 99.60 | 2585.89 | B.1 |
| H29 | Asymptomatic | Q | 31 | Y | CAMB-80AFB | 99.60 | 2028.31 | B.1 |
| H42 | Asymptomatic | Q | 35 | N | | | | Not available |
| H44 | Asymptomatic | Q | 28 | N | | | | Not available |
| H48 | Asymptomatic | Q | 36 | N | | | | Not available |
| H49 | Asymptomatic | Q | 35 | N | | | | Not available |
| H4 | Symptomatic | R | 24 | Y | CAMB-7C0D2 | 98.74 | 2083.89 | B.1 |
| H9 | Symptomatic | R | 19 | Y | CAMB-7FB65 | 99.61 | 3288.11 | B.1 |
| H13 | Symptomatic | R | 21 | Y | CAMB-7FBA1 | 99.60 | 3307.61 | B.1 |
| H27 | Asymptomatic | R | 25 | Y | CAMB-7FCBD | 98.61 | 1085.78 | B.1 |
| H34 | Symptomatic | R | 30 | Y | CAMB-81025 | 99.60 | 1997.98 | B.1 |
| H37 | Asymptomatic | R | 35 | N | | | | Not available |
| H52 | Asymptomatic | R | 34 | | | | | |
| H58 | Symptomatic | R/S/A/Q/P/L/N/M/K/Other | 24 | | | | | |
| H15 | Symptomatic | S/N | 32 | Y | CAMB-7FBDE | 99.60 | 2246.43 | B.1.7 |
| H41 | Asymptomatic | S/Q | 31 | N | | | | Not available |
| H2 | Asymptomatic | T | 36 | Y | CAMB-7C0B4 | 93.55 | 293.223 | B.1 |
| H26 | Asymptomatic | T | 32 | Y | CAMB-7FCAE | *0.03* | *0.189437* | *Not available* |
| H50 | Symptomatic | T | 34 | N | | | | Not available |
| H43 | Asymptomatic | U | 32 | N | | | | Not available |

route of entry into the programme, the process for testing and follow-up was identical. Wards were closed to external visitors.

We devised a scoring system to determine the clinical probability of COVID-19 based on symptoms from existing literature (*Wang et al., 2020*; *Giacomelli et al., 2020*; *Table 1*). Self-referring HCW and staff captured by daily workforce sickness reports were triaged by designated Occupational Health nurses using these criteria (*Table 2*). Self-isolating staff in the medium and low probability categories were prioritised for testing, since a change in the clinical management was most likely to derive from results.

Self-isolation and household quarantine advice was determined by estimating the pre-test probability of COVID-19 (high, medium or low) in those with symptoms, based on the presence or absence of typical features (*Tables 1–2*). Symptom history was obtained for all symptomatic HCWs at the time of self-referral, and again for all positive cases via telephone interview when results became available. All individuals who had no symptoms at the time of testing were followed up by telephone within 14 days of their result. Pauci-symptomatic individuals were defined as those with low-probability clinical COVID-19 criteria (*Table 2*).

## Sample collection procedures

Testing was primarily undertaken at temporary on-site facilities. Two 'Pods' (self-contained portable cabins with office, kitchen facilities, generator and toilet) were erected in close proximity both to the laboratory and main hospital. Outside space was designed to enable car and pedestrian access, and ensure ≥2 m social distancing at all times. Individuals attending on foot were given pre-prepared self-swabbing kits containing a swab, electronically labelled specimen tube, gloves and swabbing instructions contained in a zip-locked collection bag. Pods were staffed by a team of re-deployed research nurses, who facilitated self-swabbing by providing instruction as required. Scenario 1 PPE (*Table 5*) was worn by Pod nurses at all times. Individuals in cars were handed self-swabbing kits through the window, with samples dropped in collection bags into collection bins outside. Any children (household contacts) were brought to the pods in cars and swabbed in situ by a parent or guardian.

In addition to Pod-based testing, an outreach HCW asymptomatic screening service was developed to enable self-swabbing kits to be delivered to HCWs in their area of work, minimising disruption to the working routine of hospital staff, and maximising Pod availability for symptomatic staff. Lists of all staff working in target areas over a 24 hr period were assembled, and kits pre-prepared accordingly. Self-swabbing kits were delivered to target areas by research nurses, who trained senior nurses in the area to instruct other colleagues on safe self-swabbing technique. Kits were left in target areas for 24 hr to capture a full cycle of shift patterns, and all kits and delivery equipment were thoroughly decontaminated with 70% ethanol prior to collection. Twice daily, specimens were delivered to the laboratory for processing.

## Laboratory methods

The swabbing, extraction and amplification methods for this study follow a recently validated procedure (*Sridhar et al., 2020*). Individuals performed a self-swab at the back of the throat followed by the nasal cavity as previously described (*Our World in Data, 2020*). The single dry sterile swab was immediately placed into transport medium/lysis buffer containing 4M guanidine thiocyanate to

**Table 7.** Distribution of positive SARS-CoV-2 tests amongst symptomatic individuals with a positive test result, categorised according to test group and COVID-19 symptom-based probability criteria (as defined in *Table 2*).

| | Distribution of COVID-19 clinical probability scores for individuals with a positive SARS-CoV-2 test result | | | |
|---|---|---|---|---|
| | High | Medium | Low | Total |
| HCW symptomatic screening group | 22/26 (85%) | 3/26 (11%) | 1/26 (4%) | 26/26 (100%) |
| HCW symptomatic household contacts | 3/4 (75%) | 0/4 (0%) | 1/4 (25%) | 4/4 (100%) |

inactivate virus, and carrier RNA. This facilitated BSL2-based manual extraction of viral RNA in the presence of MS2 bacteriophage amplification control. Use of these reagents and components avoided the need for nationally employed testing kits. Real-time RT-PCR amplification was performed as previously described and results validated by confirmation of FAM amplification of the appropriate controls with threshold cycle (CT) ≤36. Lower CT values correspond to earlier detection of the viral RNA in the RT-PCR process, corresponding with a higher copy number of the viral genome. In 2/1,270 cases, RT-PCR failed to amplify the internal control and results were discarded, with HCW offered a re-test. Sequencing of positive samples was attempted on samples with a CT ≤30 using a multiplex PCR based approach (*Quick et al., 2017*) using the modified ARTIC v2 protocol (*Quick, 2020*) and v3 primer set (*Artic network, 2020*). Genomes were assembled using reference based assembly and the bioinformatic pipeline as described (*Quick et al., 2017*) using a 20x minimum coverage as a cut-off for any region of the genome and a 50.1% cut-off for calling of single nucleotide polymorphisms (SNPs). Samples were sequenced as part of the COVID-19 Genomics UK Consortium, COG-UK), a partnership of NHS organisations, academic institutions, UK public health agencies and the Wellcome Sanger Institute.

## Results reporting

As soon as they were available, positive results were telephoned to patients by Infectious Diseases physicians, who took further details of symptomatology including timing of onset, and gave clinical advice (*Table 2*). Negative results were reported by Occupational Health nurses via telephone, or emailed through a secure internal email system. Advice on returning to work was given as described in *Table 2*. Individuals advised to self-isolate were instructed to do so in their usual place of residence. Particularly vulnerable staff or those who had more severe illness but did not require hospitalisation were offered follow-up telephone consultations. Individuals without symptoms at the time of testing were similarly followed up, to monitor for de novo symptoms. Verbal consent was gained for all results to be reported to the hospital's infection control and health and safety teams, and to Public Health England, who received all positive and negative results as part of a daily reporting stream.

## Data extraction and analysis

Swab result data were extracted directly from the hospital-laboratory interface software, Epic (Verona, Wisconsin, USA). Details of symptoms recorded at the time of telephone consultation were extracted manually from review of Epic clinical records. Data were collated using Microsoft Excel, and figures produced with GraphPad Prism (GraphPad Software, La Jolla, California, USA). Fisher's exact test was used for comparison of positive rates between groups defined in the main text. Mann-Whitney testing was used to compare CT values between different categories of tested individuals. HCW samples that gave SARS CoV-2 genomes were assigned global lineages defined by *Rambaut et al., 2020* using the PANGOLIN utility (*O'Toole and McCrone, 2020*).

## Ethics and consent

As a study of healthcare-associated infections, this investigation is exempt from requiring ethical approval under Section 251 of the NHS Act 2006 (see also the NHS Health Research Authority algorithm, available at http://www.hra-decisiontools.org.uk/research/, which concludes that no formal ethical approval is required). Written consent was obtained from each HCW described in the anonymised case vignettes.

## Acknowledgements

This work was supported by the Wellcome Trust Senior Research Fellowships 108070/Z/15/Z to MPW, 215515/Z/19/Z to SGB and 207498/Z/17/Z to IGG; Collaborative award 206298/B/17/Z to IGG; Principal Research Fellowship 210688/Z/18/Z to PJL; Investigator Award 200871/Z/16/Z to KGCS; Addenbrooke's Charitable Trust (to MPW, SGB, IGG and PJL); the Medical Research Council (CSF MR/P008801/1 to NJM); NHS Blood and Transfusion (WPA15-02 to NJM); National Institute for Health Research (Cambridge Biomedical Research Centre at CUHNFT), to JRB, MET, AC and GD, Academy of Medical Sciences and the Health Foundation (Clinician Scientist Fellowship to MET), Engineering and Physical Sciences Research Council (EP/P031447/1 and EP/N031938/1 to RS),Cancer Research UK (PRECISION Grand Challenge C38317/A24043 award to JY). Components of this

work were supported by the COVID-19 Genomics UK Consortium, (COG-UK), which is supported by funding from the Medical Research Council (MRC) part of UK Research and Innovation (UKRI), the National Institute of Health Research (NIHR) and Genome Research Limited, operating as the Wellcome Sanger Institute

**The CITIID-NIHR COVID-19 BioResource Collaboration**

**Principal Investigators**: Stephen Baker, John Bradley, Gordon Dougan, Ian Goodfellow, Ravi Gupta, Paul J Lehner, Paul A Lyons, Nicholas J Matheson, Kenneth GC Smith, M Estee Torok, Mark Toshner, Michael P Weekes

**Infectious Diseases Department:** Nicholas K Jones, Lucy Rivett, Matthew Routledge, Dominic Sparkes, Ben Warne

**SARS-CoV-2 testing team:** Josefin Bartholdson Scott, Claire Cormie, Sally Forrest, Harmeet Gill, Iain Kean, Mailis Maes, Joana Pereira-Dias, Nicola Reynolds, Sushmita Sridhar, Michelle Wantoch, Jamie Young

**COG-UK Cambridge Sequencing Team:** Sarah Caddy, Laura Caller, Theresa Feltwell, Grant Hall, William Hamilton, Myra Hosmillo, Charlotte Houldcroft, Aminu Jahun, Fahad Khokhar, Luke Meredith, Anna Yakovleva

**NIHR BioResource:** Helen Butcher, Daniela Caputo, Debra Clapham-Riley, Helen Dolling, Anita Furlong, Barbara Graves, Emma Le Gresley, Nathalie Kingston, Sofia Papadia, Hannah Stark, Kathleen E. Stirrups, Jennifer Webster

**Research nurses:** Joanna Calder, Julie Harris, Sarah Hewitt, Jane Kennet, Anne Meadows, Rebecca Rastall, Criona O,Brien, Jo Price, Cherry Publico, Jane Rowlands, Valentina Ruffolo, Hugo Tordesillas

**NIHR Cambridge Clinical Research Facility:** Karen Brookes, Laura Canna, Isabel Cruz, Katie Dempsey, Anne Elmer, Naidine Escoffery, Stewart Fuller, Heather Jones, Carla Ribeiro, Caroline Saunders, Angela Wright

**Cambridge Cancer Trial Centre:** Rutendo Nyagumbo, Anne Roberts

**Clinical Research Network Eastern:** Ashlea Bucke, Simone Hargreaves, Danielle Johnson, Aileen Narcorda, Debbie Read, Christian Sparke, Lucy Warboys

**Administrative staff, CUH:** Kirsty Lagadu, Lenette Mactavous

**CUH NHS Foundation Trus**t: Tim Gould, Tim Raine, Ashley Shaw

**Cambridge Cancer Trials Centre:** Claire Mather, Nicola Ramenatte, Anne-Laure Vallier

**Legal/Ethics:** Mary Kasanicki

**CUH Improvement and Transformation Team:** Penelope-Jane Eames, Chris McNicholas, Lisa Thake

**Clinical Microbiology & Public Health Laboratory (PHE):** Neil Bartholomew, Nick Brown, Martin Curran, Surendra Parmar, Hongyi Zhang

**Occupational Health:** Ailsa Bowring, Mark Ferris, Geraldine Martell, Natalie Quinnell, Giles Wright, Jo Wright

**Health and Safety:** Helen Murphy

**Department of Medicine Sample Logistics:** Benjamin J. Dunmore, Ekaterina Legchenko, Stefan Gräf, Christopher Huang, Josh Hodgson, Kelvin Hunter, Jennifer Martin, Federica Mescia, Ciara O'Donnell, Linda Pointon, Joy Shih, Rachel Sutcliffe, Tobias Tilly, Zhen Tong, Carmen Treacy, Jennifer Wood

**Department of Medicine Sample Processing and Acquisition:** Laura Bergamaschi, Ariana Betancourt, Georgie Bowyer, Aloka De Sa, Maddie Epping, Andrew Hinch, Oisin Huhn, Isobel Jarvis, Daniel Lewis, Joe Marsden, Simon McCallum, Francescsa Nice, Ommar Omarjee, Marianne Perera, Nika Romashova, Mateusz Strezlecki, Natalia Savoinykh Yarkoni, Lori Turner

**Epic team/other computing support:** Barrie Bailey, Afzal Chaudhry, Rachel Doughton, Chris Workman

**Statistics/modelling**: Richard J Samworth, Caroline Trotter

## Additional information

### Group author details

**The CITIID-NIHR COVID-19 BioResource Collaboration**

Ravi Gupta; **Paul A Lyons; Mark Toshner; Ben Warne**; Josefin Bartholdson Scott; Claire Cormie; Harmeet Gill; Iain Kean; Mailis Maes; Nicola Reynolds; Michelle Wantoch; Sarah Caddy; Laura Caller; Theresa Feltwell; Grant Hall; Myra Hosmillo; Charlotte Houldcroft; Aminu Jahun; Fahad Khokhar; Anna Yakovleva; Helen Butcher; Daniela Caputo; Debra Clapham-Riley; Helen Dolling; Anita Furlong; Barbara Graves; Emma Le Gresley; Nathalie Kingston; Sofia Papadia; Hannah Stark; Kathleen E Stirrups; Jennifer Webster; Joanna Calder ; Julie Harris; Sarah Hewitt; Jane Kennet ; Anne Meadows; Rebecca Rastall; Criona O Brien; Jo Price; Cherry Publico; Jane Rowlands; Valentina Ruffolo; Hugo Tordesillas; Karen Brookes; Laura Canna; Isabel Cruz; Katie Dempsey; Anne Elmer; Naidine Escoffery; Heather Jones; Carla Ribeiro; Caroline Saunders; Angela Wright; Rutendo Nyagumbo; Anne Roberts; Ashlea Bucke; Simone Hargreaves; Danielle Johnson; Aileen Narcorda; Debbie Read; Christian Sparke; Lucy Warboys; Kirsty  Lagadu; Lenette Mactavous; Tim Gould; Tim Raine; Claire Mather; Nicola Ramenatte; Anne-Laure Vallier; Mary Kasanicki; Penelope-Jane Eames; Chris McNicholas; Lisa Thake; Neil Bartholomew; Nick Brown; Surendra  Parmar ; Hongyi Zhang; Ailsa Bowring; Geraldine Martell; Natalie Quinnell; Jo Wright; Helen Murphy; Benjamin J Dunmore; Ekaterina Legchenko; Stefan Gräf; Christopher Huang; Josh Hodgson; Kelvin Hunter; Jennifer Martin; Federica Mescia; Ciara  O'Donnell; Linda Pointon; Joy Shih; Rachel Sutcliffe; Tobias Tilly; Zhen Tong; Carmen Treacy ; Jennifer Wood; Laura Bergamaschi; Ariana Betancourt; Georgie Bowyer; Aloka De Sa; Maddie Epping; Andrew Hinch; Oisin Huhn; Isobel Jarvis; Daniel Lewis; Joe Marsden; Simon McCallum; Francescsa Nice

### Competing interests

M Estee Torok: Reports grants from Academy of Medical Sciences and the Health Foundation, non-financial support from National Institute of Health Research, grants from Medical Research Council, grants from Global Challenges Research Fund, personal fees from Wellcome Sanger Institute, personal fees from University of Cambridge, personal fees from Oxford University Press. Afzal Chaudhry: Reports grants from Cambridge Biomedical Research Centre at CUHNFT. Richard J Samworth: Reports grants from EPSRC fellowship. Gordon Dougan: Reports grants from NIHR. Kenneth GC Smith, Michael P Weekes: Reports grants from Wellcome Trust. Paul J Lehner, Ian G Goodfellow, Stephen Baker: Reports grants from Wellcome Trust and Addenbrooke's Charitable Trust. Nicholas J Matheson: Reports grants from MRC (UK) and NHS Blood and Transfusion. The other authors declare that no competing interests exist.

### Funding

| Funder | Grant reference number | Author |
| --- | --- | --- |
| Wellcome | 108070/Z/15/Z | Michael P Weekes |
| Wellcome | 215515/Z/19/Z | Stephen Baker |
| Wellcome | 207498?Z/17/Z | Ian G Goodfellow |
| Wellcome | 206298/B/17/Z | Ian G Goodfellow |
| Wellcome | 210688/Z/18/Z | Paul J Lehner |
| Wellcome | 200871/Z/16/Z | Kenneth G C Smith |
| Addenbrooke's Charitable Trust, Cambridge University Hospitals | | Paul J Lehner<br>Ian G Goodfellow<br>Stephen Baker<br>Michael P Weekes |
| Medical Research Council | MR/P008801/1 | Nicholas J Matheson |
| NHS Blood and Transplant | WPA15-02 | Nicholas J Matheson |

| National Institute for Health Research | Cambridge Biomedical Research Centre | John R Bradley<br>M Estee Torok<br>Afzal Chaudhry<br>Gordon Dougan |
|---|---|---|
| Academy of Medical Sciences | Clinician Scientist Fellowship | M Estee Torok |
| Engineering and Physical Sciences Research Council | EP/P031447/1 | Richard J Samworth |
| Engineering and Physical Sciences Research Council | EP/N031938/1 | Richard J Samworth |
| Cancer Research UK | PRECISION Grand Challenge C38317/A24043 | Jamie Young |

The funders had no role in study design, data collection and interpretation, or the decision to submit the work for publication.

### Author contributions

Lucy Rivett, Conceptualization, Data curation, Formal analysis, Investigation, Methodology, Project administration, Writing - review and editing; Sushmita Sridhar, Conceptualization, Data curation, Formal analysis, Validation, Methodology, Project administration, Writing - review and editing; Dominic Sparkes, Data curation, Formal analysis, Writing - original draft, Project administration, Writing - review and editing; Matthew Routledge, Writing - original draft, Project administration, Writing - review and editing; Nick K Jones, Data curation, Investigation, Methodology, Writing - original draft, Project administration, Writing - review and editing; Sally Forrest, Data curation, Validation, Project administration; Jamie Young, Data curation, Formal analysis, Investigation; Joana Pereira-Dias, Luke Meredith, Richard J Samworth, Data curation, Formal analysis; William L Hamilton, Data curation, Writing - original draft; Mark Ferris, Conceptualization, Writing - original draft, Project administration, Writing - review and editing; M Estee Torok, Data curation, Supervision, Writing - review and editing; The CITIID-NIHR COVID-19 BioResource Collaboration, Conceptualization, Data curation, Formal analysis, Funding acquisition, Investigation, Writing - original draft, Project administration; Martin D Curran, Conceptualization, Methodology, Project administration; Stewart Fuller, Project administration; Afzal Chaudhry, Data curation, Software; Ashley Shaw, Supervision, Project administration; John R Bradley, Gordon Dougan, Kenneth GC Smith, Paul J Lehner, Nicholas J Matheson, Supervision, Project administration, Writing - review and editing; Giles Wright, Project administration, Writing - review and editing; Ian G Goodfellow, Data curation, Formal analysis, Supervision, Project administration, Writing - review and editing; Stephen Baker, Conceptualization, Data curation, Formal analysis, Methodology, Writing - original draft, Project administration, Writing - review and editing; Michael P Weekes, Conceptualization, Data curation, Investigation, Methodology, Writing - original draft, Project administration, Writing - review and editing

### Author ORCIDs

Lucy Rivett ⬤ https://orcid.org/0000-0002-2781-9345
Paul J Lehner ⬤ https://orcid.org/0000-0001-9383-1054
Nicholas J Matheson ⬤ https://orcid.org/0000-0002-3318-1851
Ian G Goodfellow ⬤ https://orcid.org/0000-0002-9483-510X
Michael P Weekes ⬤ https://orcid.org/0000-0003-3196-5545

### Ethics

Human subjects: As a study of healthcare-associated infections, this investigation is exempt from requiring ethical approval under Section 251 of the NHS Act 2006 (see also the NHS Health Research Authority algorithm, available at http://www.hra-decisiontools.org.uk/research/, which concludes that no formal ethical approval is required). Written consent was obtained from each HCW described in the anonymised case vignettes.

### Decision letter and Author response

Decision letter https://doi.org/10.7554/eLife.58728.sa1

Author response https://doi.org/10.7554/eLife.58728.sa2

## Additional files

### Supplementary files
- Source data 1. Asymptomatic SARS-CoV-2 screening programme source data.
- Transparent reporting form

### Data availability
Sequencing data have been deposited in GSAID under accession codes EPI_ISL_433989-EPI_-ISL_433992, EPI_ISL_434005, EPI_ISL_433489-EPI_ISL_433497. Researchers will be prompted to register and log on to the website to access the datasets (https://www.epicov.org/epi3/frontend#1f1442).

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
