## [Decision Letter]

**Acceptance summary:**

Your paper nicely demonstrates the importance of systematic and comprehensive testing of coronavirus infection in healthcare workers. Asymptomatic carriers, albeit a minority, are detected this way, and their contribution to transmission of the virus in the hospital setting can be estimated.

**Decision letter:**

Your submission has been evaluated by a Deputy Editor and a group of Senior Editors, and they considered your work would qualify for fast track. The paper was assigned to me to judge the quality of the manuscript, of the reviews, and the revisions carried out based on the reviews. I must say that I was impressed by the quality and the novelty of the revised paper. Also the seven reviewers had provided very positive, constructive and complete reviews.

---

## [Author Response]

[Editors' note: we include below the reviews that the authors received from another journal, along with the authors’ responses.]

Comments to the author:

Reviewer #1:1) Section “Research in context”: Abstracts were screened and judged for relevance, but there is no indication of the criteria for this relevance assessment. This should be described.

Many thanks for pointing out this omission. Amended, the line now reads “Abstracts were manually screened and judged for potential relevance. If the manuscript discussed HCW testing for SARS-CoV-2 the contents of the paper were reviewed and, where appropriate, referenced.”

2) Please define HCW, does this include people such as chaplains, for example?

Thank you for querying this. We have now provided a definition of HCW for the purposes of this study in the first paragraph of the methods section of the revised manuscript to improve clarity.

3) What about the other hospital staff who are also near patients but not giving treatment/care? For example, cleaners, porters, and people delivering meal trays? They could also potentially transmit the virus to patients… why was the study limited to HCWs? Later in the paper you use the term "patient-facing", better to get terms all organized earlier in the paper (see my comment 2 above).

Please see our response to point 2 for clarification of the range of HCW that were offered testing. No symptomatic patient-facing or non-patient-facing staff were excluded from testing. No patient-facing or non-patient facing staff working in areas targeted by the asymptomatic screening arm of the programme were excluded from testing. We have altered the wording of the discussion to avoid confusion with regards to this.

4) Abstract methods: "throat/nose self-swab" is not clear and it seems the "/" should be a "+"

We have corrected this as suggested.

5) The methods should be clearer to state that household contacts included adults and children (I had to hunt for the latter detail in the supplement).

We have corrected this as suggested.

6) Are you stating that written informed consent was only obtained from people in the case vignettes rather than everyone in the study? Even if research ethics committee approval was not required for this study in the UK, it would be ETHICAL to get written informed consent from all participants because HCWs are a vulnerable population. Also, you need this formal consent process in order to explain to them the details about data sharing, a huge issue because the UK is still following the GDPR until Dec 31st.

This was run as a service evaluation through occupational health, HCW agreed to be tested. No personal identifiable data has been shared or disclosed so GDPR does not apply.

7) For self-isolation, were people sent home or somewhere else?

Individuals were requested to self-isolate in their usual residence. We have added this information for clarity to the main text.(See SI, paragraph “Results reporting”)

8) What do you mean by "PPE reviews were undertaken"? Was this to ask them if they had PPE, enough PPE, the proper PPE, if they were re-using PPE? If you found troubling answers, did you alert anyone?

We have elaborated on the nature of PPE reviews and interventions enacted in the event of improper PPE use in the revised manuscript. No instances of inadequate provision of equipment were observed, so training and education on correct use of the equipment provided was the only means of intervening.

9) This sentence is confusing: "Given the low seasonal incidence of alternative viral causes of high COVID-19 probability symptoms in the UK26, and the high potential for SARS-CoV-2 hospital exposure during the pandemic, we took the pragmatic approach to allow continued work by these individuals, as long as they had been entirely well for the preceding 48 hours." But these were people who you knew were positive? 48hrs of no symptoms is not enough? Shouldn't they be negative before you allow them to work? Symptoms can wax and wane.

Many thanks for pointing out the clarity needed here. This is a key issue with testing currently asymptomatic HCWs. Given the current evidence suggests that viable, infectious virus is rarely shed longer than 8 days post-onset of symptoms (Wolfel et al., 2020) and that viral RNA can continue to be shed for up to 37 days and probably beyond (Zhou et al., 2020), we took the approach that these HCWs (provided they were currently well) did not pose a significant infectious risk and could continue to work. In essence, we felt that a positive test in this group should not cause us to deviate from the current symptom-based isolation advice. The exception to this rule concerned those who worked with bone marrow transplant recipients, where we followed recent national guidance requiring a negative test prior to a return to work.

Given the long period of post-symptomatic RNA shedding, as screening of asymptomatic HCWs is expanded, clear, national, guidelines will be required to deal with this scenario. Prolonged isolation of HCWs, possibly for many weeks after their illness will have a major impact on the workforce, potentially with no benefit towards infection control.

For clarity, the sentence now reads:

“This approach was based on the following: low seasonal incidence of alternative viral causes of high COVID-19 probability symptoms in the UK26, the high potential for SARSCoV-2 exposure during the pandemic and the potential for prolonged, non-infectious shedding of viral RNA.”

10) Regarding Table S4, tell us more clearly if there was a connection re PPE scenario and your clinical findings. Also, refer back to my comment 8, maybe the scenario did not actually match to the actuality of PPE access/use in the ward (the PPE review)? I need more clarity here. I only see a brief mention about Ward Q for Figure 3.

Thank you for this suggestion. We have now added a more detailed description of the hospital’s routine COVID-19 infection control and PPE practices as a caption to Table

S4 of the revised manuscript. We have also added more detail about ward Q (See Table S2 (previously Table S4) and Results section).

11) Did any of the wards allow external/community visitors?

A line has been added to the methods section highlighting that wards were closed to external visitors.

Reviewer #2:This is a timely and important manuscript reporting a SARS-CoV-2 surveillance program in healthcare workers (HCWs) at a teaching hospital in the UK. It is very important that they report test-positive HCWs that are not showing clinical symptoms. These results should be considered by all jurisdictions that are designing and implementing SARS-CoV-2 testing protocols for HCWs.

Many thanks to this reviewer for the complimentary assessment of our work.

1) Overarching comment: it is unfortunate that the authors did not appear to investigate exposures of the positive HCW cases in this study. It appears that they conducted telephone interviews to record information on clinical symptoms, but not potential exposures. Without exposure information, it is difficult to know the degree to which positive HCWs were infected at work versus from community or household exposure. Preliminary data reported by the WHO from China suggests that the majority of HCW cases may be from household, not occupational exposure. A recent study from the US CDC suggests a higher proportion of cases in HCWs were in individuals that only had known exposure to SARS-CoV-2-postive cases, but it is only a small portion of HCW cases that may not be representative of the broader US situation.

Many thanks for this important point. However, in order to draw strong conclusions about the predominant sources of HCW infection, an entirely new study would be required. This would be likely to require visits to and sampling of individual households, taking detailed epidemiological data about all members of the household, recurrent visits to follow up development of symptoms and sequencing-based studies of any positive household isolates. In Cambridge UK, many HCWs live in shared accommodation, often with multiple other HCW. This can further complicate conclusions as to work vs household acquisition of infection. An additional concern is that given the wide spectrum of clinical features from asymptomatic – paucisymptomatic – truly symptomatic, even with extremely detailed epidemiological information and sequencing, the direction of transmission may still be difficult to disentangle.

Anecdotally, several HCWs did mention household contacts who also had symptoms or who had previously tested positive for SARS-CoV2. We have included some of this data in the case vignettes for illustrative purposes (see Table 2).

2) I feel this study missed this opportunity to collect these exposure data from a captive patient population. It is not a fatal flaw, just an unfortunate circumstance. They do highlight the future need for this work in the discussion. The study claims to have identified SARS-CoV-2 transmission within the hospital, and the results, particularly sequencing analysis, support this to some degree. However, the story would be stronger with epidemiologic reporting of exposure information of positive HCWs. I think this limitation needs to be more clearly stated in the discussion. The authors rightly state that strain identification through sequencing is not a discriminatory method to pin-point viral sources of infection, given the relatively low mutation rate of coronaviruses. It is not clear from the study design/reported methods if tested household contacts were the source of or recipient of the virus to/from the positive HCWs. This is another related piece of information that could have been teased out through epidemiologic investigation.

Thank you for this suggestion. We have added to the discussion to highlight to this study limitation.

3) Abstract Methods: please include the location of the hospital (city, country) to direct the reader to the population of interest for the study beyond HCWs in a teaching hospital.

We have added these details.

4) Abstract Findings: is there room to include the sequencing analysis of the symptomatic HCW screening group and symptomatic household contacts?

Many thanks for this suggestion. Unfortunately due to limits on the word count in the Abstract, and our concern to emphasise the nuanced nature of interpretation of the sequencing data, we decided to limit discussion of this data in the Abstract. We state:

“Viral genome sequencing showed that the majority of HCWs had the dominant lineage B.1.”

We discuss this data in more detail in the remainder of the manuscript.

5) Methods: please clarify how the two parallel streams of testing are integrated. The results make reference to "RT-PCR failure rate" that is presumably tied to this, but it is never explained (main text or supplementary info).

Many thanks for this suggestion; we have updated the methods section as requested to describe entry into and subsequent integration of the two parallel streams of testing

(See “Staff screening protocols”). We also define “RT-PCR failure” as “failed internal control”.

6) There are a number of pieces of "Methods" presented in the results – some specific examples listed below.

Many thanks for pointing these out; we have addressed all concerns below

7) Methods/Results: risk of anticipated exposure to COVID-19 (red, amber, green) – definitions of what constitutes a high, medium and low risk ward should be provided in the main text (of the Methods section, not results), or at least referred to Table S3 in the Supplementary Materials. The sentence in Staff Screening Protocols alludes to these risk areas, but this must be made explicit to the reader. It would also be nice if Table S3 could provide concrete examples of types of wards/hospital service areas that fell into the different criteria. I realize that part of the criteria consider whether or not the area has held a confirmed SARS-CoV-2, but presumably some types of wards/services are always at higher risk and would play into this categorization. Table S4 gets more into these details, but still is not a concrete description – just general terms like "areas where aerosol-generating procedures are carried out". The reason I feel this is important is to provide some guidance on the types of hospital wards/services that are at higher risk, not just those called "red" or "amber" based on the definitions. Maybe this is better provided in the descriptive results – the types of wards/services falling into each risk category based on the definitions.

We have now included examples of the service areas that fall into red/amber/green categories in Table S1 (previous Table S3), since this table additionally includes details of PPE used in each area. We have explicitly referenced Tables S1-S2

(previously Table S3-S4) in the Methods section as requested.

8) Results – please define what is meant by "RT-PCR failure rate" – it is not clear to the reader – please define in the methods. This might relate to the parallel streams of testing, but this is not clear from the main text and must be explained.

As detailed above, we have now explained this in the methods section.

9) Results text referring to Figure 1—figure supplement 1: please provide the linear regression coefficient and P-value for this lack of correlation between CT values and days since symptom onset in this analysis. Please provide a brief description of the linear regression in the Supplementary Info (how was the linearity of a continuous predictor checked with the outcome? How were model assumptions checked?

Many thanks for this suggestion. We note that reviewer 6 also asked about the appropriateness of the use of a linear regression model to analyse these data. Reviewer 4 asked for further details about the relationship between symptom group (asymptomatic/early symptomatic/late-post symptomatic) and Ct values.

We agree that the underlying assumptions of normality in the linear regression model may not apply to our data. A better approach would be a non-parametric analysis of between-group differences. (see Figure 1—figure supplement 1).

We have therefore re-analysed the data originally presented in Figure 1—figure supplement 1, grouping positive individuals according to days since symptom onset in line with the definitions we used in the text (symptomatic ≤7 days, symptomatic >7 days). We have presented these data using a similar layout to Figure 1. Between the early- and late- symptomatic groups we did not find a relationship between symptom duration and Ct value using nonparametric testing (Mann-Whitney test).

Note that this reanalysis does not change our original findings, of a lack of a clear relationship between Ct value and symptom duration.

The reviewer may also wish to note our response to reviewer 4 who asked us to assess any relationship between our symptom-based risk classification (Low/Medium/High risk) and Ct value for individuals testing positive. We were unable to detect a difference between risk groups, but were underpowered to test for this due to the relatively small number of individuals in the medium/low risk groups. We have made an adjustment to the manuscript to mention this, without over-interpreting these findings.

10) Results section (Three subgroups of…): the sentences presenting statistical results in the last third of this paragraph are very confusing. I had great difficulty interpreting them. They could be better presented in a Table.

We agree with the reviewer that these findings were reported in a confusing manner. We have simplified the description of these data in the text and improved the crossreferencing to Figure 2.

11) This links to comments on Figure 3, which would be better to include in this table than a figure. It would allow for inclusion of 95% CIs, which I strongly recommend, and would avoid the confusion of the number of tests scale from Figure 3 that makes it look like most wards has zero test positives. It also avoids any issue with red-green colour blindness.

We thank the reviewer for this suggestion. For clarity, we have added a supplementary table (now Table S5) that also shows the data from this figure.

12) Results: define PPE for the first time.

We have corrected this omission.

13) Results: what kind of ward was Ward Q?

Thank you for querying this. We have provided further detail on the nature of ward Q in the revised manuscript.

14) Table 1: title caption should be changed to indicate that these results include total number of tests performed and number/proportions of positive results. These proportions should include measures of dispersion (preferably a 95% confidence intervals).

We thank the reviewer for this suggestion, and have amended the caption for Table 1.

However, we do not agree that the proportions should include measures of dispersion – this table gives a descriptive account of the number/proportions of positive results we observed. Please also see our reply to reviewer 6, where we justify our use of Fisher’s exact test as opposed to the use of confidence intervals for binomial proportions.(see

Table 1)

15) Table 2: clinical vignettes. While these are somewhat interesting, they could provide more information, particularly about possible SARS-CoV-2 exposures. For each one, I wanted to know what their occupational versus home/community exposure risks were. Without this, I don't find this section as informative. It appears from the study that detailed exposure information was not sought from the study participants. This is an unfortunate omission as this information is vitally important to understand the true occupational transmission of SARS-CoV-2 to HCWs. The sequence analysis suggests commonality, but also variability in that Ward F has multiple viral strains.

Many thanks for this suggestion. We have revised this section to make it more informative in general (as also detailed above in response to “overarching comment”), in particular giving more detail about potential exposures (see Table 2).

16) Table 3: positive tests by symptom-based probability. The title of this table should refer the reader to Table S2 for the definitions of the clinical probability categories. This table should also include measures of dispersion (95% CIs).

We have adjusted the title of this table to refer to Table S4 (previously S2) as suggested. Since the proportions given show the distribution of the *positive* test results within each group according to clinical risk group and not the percentage of *all* test results for each screening group (i.e. note that the row totals sum to 100%) we feel it is not appropriate to apply measures of dispersion. We have amended the figure legend to clarify this.

17) Figure 1: CT should be defined.

We have corrected this omission.

18) Figure 2: Please clarify this figure – probably better presented as a table rather than the pie charts. This would allow inclusion of 95% CIs and avoid any red/green colour confusion for anyone with color blindness. Further, the info presented in the center pie chart is described will in the text (including the categorization), but is not well-described in the figure caption (and it is confusing). If kept as a pie chart (not my preference), it should be made clear that the left and right pie charts for the proportions in high/medium/low are for either ii or iii, respectively, with labels or description in the figure. Second, what to the proportions in the center pie chart represent? It looks like the proportion of all positive tests.

Many thanks for this suggestion. We have updated the colours in the pie chart to avoid difficulty for anyone with colour blindness. We have removed the confusing information about the central pie chart and referenced the main text, and clarified other information highlighted above in the figure legend. We did not think it appropriate to add 95% CIs, since the numbers described are actual numbers of HCW. (See Figure 2 and Results section.)

19) Figure 4: what is the scale on the Y-axis – number of positive tests? What is the "reactive screening group"? This term is partially defined in the Results but should be defined in the figure for clarity.

We have amended the Y axis label on Figure 4 to clarify that this is the number of tests returning positive for SARS-CoV-2. In line with the reviewer recommendation, we have expanded the figure legend to clarify the meaning of each group.(see Figure 4)

20) Supplementary Material: Tables S1, S2, S3 and S4 need titles to better orient the reader. They are referred to and are important to the case and risk definitions in the main text. One cannot go to the tables and understand what they include without Table titles without inferring it from the text. For example, Table S1 refers to Major and Minor symptoms, but this is not evident from looking at the table itself.

Thank you for this suggestion. We have added titles to Tables S1-S4 to improve clarity.

21) Table S3: the main text refers to the alignment of risk categorization of red = high, amber = medium, and green = low risk in the Results section. This makes sense, but in Table S3 (again, requires a title), should explicitly list that red = high, etc.

Thank you for this suggestion. We have added labels to the columns.

22) Table S5 caption: this should include details as to which isolates were included (e.g., all isolates from the study? All HCW isolates? What about household exposure isolates?).

Thank you for this suggestion. We have added updated the Table S5 caption to clarify this. Isolates from all HCWs and household contacts in the study are included in the table.

23) Figure 4—figure supplement 1: the colours are too similar to be able to distinguish the phylogenetic relationships of virus sequences from different wards. I know this is supplementary material and not in the main text, but this is very interesting and I am not able to clearly tell them apart. Maybe each could be given a number that is added to the figure, or some other way to distinguish them.

Many thanks for this helpful suggestion. We have now added ward letters to the phylogenetic tree to aid clarity. (see Figure 4—figure supplement 1)

Reviewer #3:1) Abstract: percentages are quoted to spurious accuracy. Of the 31, one could not be followed-up. This should, I suggest, be reported in Abstract and the relevant % is then 12/30 or 40%. Likewise, 75% is in fact 9/12 (say so) and provide numerator/denominator for 55% (currently reported as 54.8%). Likewise, 15% (not 15.4%) and 4/52 (8%).

We have adjusted the manuscript according to these suggestions.

2) Introduction: perhaps disconcerting that asymptomatic HCW onward transmission accounts for "only" 16-23% on top of isolation based on symptoms. Presumably depends on PPE-provision – might be worth mentioning this?

The Imperial College, London, UK report from which this estimate derives from does not specifically mention the role of PPE-provision in their model. They do however mention the importance of 24 hour turnaround for results. The sentence has been amended so it now reads “weekly testing of asymptomatic HCWs could reduce onward transmission by 1623% on top of isolation based on symptoms, provided results are available within 24 hours.”

3) Results: since 21 individuals underwent repeat testing but reasons sum to 3+11, fewer than 21. Please clarify.

We apologise for the lack of clarity here. 21 individuals underwent retesting. There was a range of reasons, including the two specifically cited in the text (evolving symptoms and “medium” clinical COVID-19 probability). The other ten individuals included those originally screened due to symptoms then re-captured as part of the asymptomatic ward screening programme. We have clarified the text as requested.

4) Results. p12, last para, sentence 1: terse – do you mean ward (green, amber, red) or household contact or what (presumably not that location of swabbing was material; but throat versus nasophyngeal could be meant – tho' I doubt it)... please rephrase.

Many thanks for spotting this omission – we have added that this means “ward”.

5) Results: I would mention FIRST that one HCW could not be contacted to obtain further history and then provide % out of 30 who were re-contacted, ie 20%, 40%, 40%.

Many thanks for this suggestion – we have updated all data and figures as suggested (throughout the paper).

6) Results, p14, para 1; NOW I'm lost as I looked at Table 1 expecting to find “green” 6/310 but I read 7/454; and for “red”, I read 20/466. WHERE do the numbers in line 3 come from – e.g., an earlier draft of paper before screening was completed?

We apologise to the reviewer for the confusion. In the text identified, we report outcomes from asymptomatic HCW screening in 21 specific clinical areas identified on the basis of risk or sickness patterns. Asymptomatic screening tests were also performed on an opportunistic basis on a small number of individuals working outside these areas, as well as in a subsequent intensified manner on ward F and ward Q after identification of clusters of positive cases on these wards.

We report the outcomes of this intensified screening programme on wards F and Q in Figure 4. In Figure 3, tests from HCWs working across >1 area are included in the statistics for each area where they worked, whereas for table 1 individual test results are aggregated according to the area of highest risk an individual HCW was exposed to. In table 1, we report all asymptomatic screen test results.

This explains the differences which the reviewer has noted. Since we recognise that this is confusing, we have added sentences of clarification the legends.

7) Results: I'd place the sentence "It was subsequently found... " at the end of the paragraph because the two persons therein confuse the reader who is trying to keep track of testable numbers, especially having already been disconcerted by inability to track in the preceding paragraph!

Many thanks, amended as suggested for improved readability.

8) Results: there were prior reasons to investigate wards F (high sickness rate and Q (PPE concerns). However, looking at Figure 3, I'd like to know more about locations O and T (both red) which had similar numbers tested as in F and Q but different % positives (~ 15% vs 5%).

Ward O was listed for further cluster investigation after the study period. We have now highlighted this in the Results section, and included a description of the ward type. Ward T was not prioritised for further cluster investigation due to the lower rate of positive tests.

9) Results: I make the count of positive tests in ward Q to be 7+1+1 = 9 but only four were sequenced. Why not the other FIVE? Please add explanation.

We have added detail that other isolates could not be sequenced due to a sample CT value >30.

10) 38.7% => 40% (12/30); data... suggest... probable symptoms (cf probability symptoms).

We thank the reviewer for noting these and have adjusted the manuscript accordingly (throughout).

11) Discussion, p17+18+19: I count 3 recommendations in main para on p18, fourth in last line of p17; two more in main para on p18; and final = 7th at top of page 18 and 8th in the next paragraph. I consider that any paper that makes 8 well-grounded and cogent recommendation from a single study has done a remarkable piece of work.

Very many thanks for this highly complementary assessment of our work.

12) Discussion might consider whether a corresponding swab-test study in patients would be warranted; or for care-home residents and staff.

Many thanks for this helpful suggestion. The sentence “Our approach has potential applications outside the hospital and may be of benefit in reducing transmission in other institutions, for example care-homes.” has been added to the closing paragraph to highlight this point.

13) Tables: Numbers don't match some cited in text.

We believe the reviewer is referring to some of the apparent discrepancies noted by reviewer 2 between the test results reported in Table 1, Figure 3, and the Results section. The reviewer may wish to note our response to reviewer 2 above in which we explain that the numbers are correctly reported, but in a manner that was initially confusing due to the different populations being studied. We have amended the text, the figure and the table legends to make this clearer. We have checked all other figures in the text and table and note that they are all in alignment.

14) Table 3. Should corresponding probability-ratings be shown for participants who tested negative? Seem to give only half the story. Answers cannot be got by subtraction between Tables 1 and 3.

We thank the reviewer for this suggestion. For those individuals who tested negative, we did not undertake the detailed symptom scoring required to assign a clinical risk group. We therefore regret that we are unable to provide this data.

15) Figure 1. The blue dots appear to be differently distributed and could almost be a mixture of two distributions. Does this notion have any subject-matter traction, eg use different blue plotting symbols for HCW (dot) and their symptomatic household contacts (square)?

We thank the reviewer for this interesting observation. We re-evaluated data in Figure 1 according to reviewer’s suggestion. A Mann-Whitney test of the differences in CT values between symptomatic household contacts and symptomatic HCWs was not significant (p=0.06), which may reflect the low number of symptomatic household contacts testing positive (n=4).

Nevertheless, further consideration of Ct values according to subgroups is clearly interesting, and we have compared values according to duration of symptoms as highlighted in our response to reviewer 2.

Reviewer #4:1) Thank you for this very important paper. We do really need data driven recommendations on screening of health care workers to reduce nosocomial infections of SARS-CoV-2. The major problem that needs to be address is that only of the 31 in your "asymptomatic" group are actually asymptomatic – the remainder are either postsymptomatic or pre-symptomatic. This needs to be much more clearly defined throughout the paper as it has major implications for infection control.

We thank the reviewer for their complimentary assessment of our work. The individuals in the HCW asymptomatic screening group were asymptomatic at the time of screening, however could be split into distinct sub-groups, with some experiencing symptoms prior to their test. We agree that this is unclear and have greatly clarified as requested, in particular in the section “Three subgroups of SARS-CoV-2 positive asymptomatic HCW”.

2) The title implies that you have built a screening program but you really have done a point prevalence survey and have not actually made recommendations as to what a screening program should look like – most importantly how often HCW should be tested. I would suggest you use your results to make recommendations on this topic or would suggest changing the title (and the objective) to reflect this as a point prevalence study to understand symptom screening and high risk wards for COVID-19 in HCW.

We respectfully disagree with the reviewer’s point. We have built a screening programme, and have described in detail how this operates in our supplementary information. The information presented is not a point prevalence survey. Rather, data were gathered over an initial 3-week period, and simply reflect the output of the programme so far. We have nevertheless changed the title as discussed above.

3) In the methods section, please define "asymptomatic" and "pauci-symptomatic" and I would recommend defining "pre-symptomatic" and "post-symptomatic" groups as well for clarity – and maintain the use of these group names throughout as your "asymptomatic" groups is not really asymptomatic – they are only without symptoms at the time of testing. Also, as these are terms that are used throughout and are important for understanding your results, please explain the schematics used for the high probability COVID symptoms and the color coding system for the wards in the text of the methods section – instead of making the reader go to the figures to understand these categorizations. Also please explain the rationale behind including the location-based designations. In the methods, you should also define Ct values and how you will use these to estimate the viral load.

Thank you for these comments. We have now defined “pauci-symptomatic” in the Methods section. We have mentioned schematics and colour coding in the methods. Location-based designations are ward locations which have been given a code so that the ward, staff and patients are not identifiable.(see Table S5). Details of CT values and relation to viral loads now in Methods.

4) In the Results section, I think it would be incredibly useful to know how many of the post-symptomatic HCW had low Ct values and culturable virus. This has incredibly important implications for return to work practices for HCW and is a point I was really hoping you would touch on. It appears that you have this data but it is not being presented in this way. Additionally, please provide data to the Results section on viral loads (either ranges of Ct values or median Ct values for each group). I think the subgroups of the asymptomatic HCW should be a major focus here as this incredibly important information for infection control programs. Additionally, please add some information in your Results section about the median amount of time that it took for HCW to perform the tests and for results to return and what your facility's specific recommendations were for those with positive results.

Many thanks for these interesting points. We were not able to collect data on culturable virus as at the time of the study we lacked the facilities to perform such analyses. We have referenced a previous study that examined this question (Wolfel et al., 2020). As described in our response to reviewer 2, we have now analysed Ct values according to subgroups. Information about time taken by HCW to perform the test was not collected, however sampling took place within a 10-minute appointment. The time from sample arrival time in the laboratory to a result being provided to the HCW was 12-36 hours, which was dependent on when each sample was taken during the working day.

5) I am a little lost in the discussion as to what your major conclusions and recommendations are based on your results. I would personally love to see Ct values for HCW by symptom groups – presymptomatic, symptomatic at the time of testing (maybe by high, medium, low clinical probability), and post-symptomatic. This would really highlight differences in infectivity and provide a great framework for the need for intermittent testing of all HCW.

Please see our response to reviewer 2, who also raised the same points.

In summary, I think this is really great work and could answer some really important questions about how all hospitals can prevent nosocomial SARS-CoV-2 infections in the future.

Many thanks again to this reviewer for their complimentary assessment of our work.

Reviewer #5:Congratulations on setting up a staff screening program in the midst of a pandemic.Major concerns:1) I found it difficult to understand whether the primary aim of your manuscript is to describe the prevalence of carriage of SARS CoV2 amongst asymptomatic healthcare workers or to describe your infection control/occupational health program. Much of the manuscript is taken up with the latter, but I found it relatively uninteresting in the absence any data related to the success (or otherwise) of what you did. I suggest you omit it and focus on your finding that truly asymptomatic carriage amongst healthcare workers is rare, even during a major pandemic.

Thank you for this suggestion. We have now described the rate of true asymptomatic infection in the Results sections of the revised manuscript and at the beginning of the Discussion. It is important to recognise, however, that the milder symptoms experienced by individuals testing positive for SARS-CoV-2 may be unrelated to COVID-19 in some instances, even if temporally related. The fact that only 15% of symptomatic HCWs tested positive for SARS-CoV-2 in our study serves as a reminder that there are many potential alternative causes of the symptoms associated with COVID-19, which may occur concurrently with SARS-CoV-2 infection. We have added a sentence to the discussion to this effect.

2) The title of the manuscript is misleading: A comprehensive healthcare worker (HCW) screening programme identifies asymptomatic HCWs as a source of nosocomial SARSCoV-2 transmission. You have identified asymptomatic carriers of SARS-CoV2 amongst healthcare workers but you have not demonstrated that they are a source of transmission.

Many thanks for this point. We have changed the title as suggested to: “A comprehensive healthcare worker SARS-CoV-2 screening programme to detect asymptomatic infection: a prospective cohort study.”

3) Please present all results in the Results section, rather than the Discussion section.

We have re-checked our manuscript, and found that the only data highlighted in the discussion that is not directly highlighted in the text of the Results section is: “57% of HCWs from asymptomatic screening group were pauci- or asymptomatic…”. However, this number is directly derivable from Figure 2.

4) Please describe the inclusion criteria for the asymptomatic screening program – what were the criteria for being defined as "staff" on those wards. Was it voluntary? What percentage of staff were actually tested?

We have added detail that testing was a voluntary process, and offered to all individuals working in a given ward during the time of sampling.

5) What were your criteria for "pauci-symptomatic"?

Please see our response to the reviewer’s first question above.

6) Please give the data for the prevalence of SARS-CoV2 positivity amongst staff who were both truly asymptomatic and had not been symptomatic previously (unless symptom onset was >24 days prior to testing). These data should also be given in the Abstract. If I have understood your results correctly it may be as low as 6/1032 (approximately 0.6%)

Thank you for this suggestion. We have added this to the Results section of the revised manuscript.

6) When reporting prevalence it is important for the reader to understand the context. For those who are not familiar with the situation in the UK (or even better in your region) at the time of your study (even those who are may not remember the situation when they read your paper in the future) please give some data on the situation in the community. I appreciate that low rate of testing in the UK makes this difficult but data such as COVID deaths per head of population will at least provide some context. You should also mention in your discussion that your prevalence (at least in green zones) may simply be the community prevalence or may be lower than the community rate. The incidence in asymptomatic pregnant women in New York City was 13.7%. (https://www.nejm.org/doi/full/10.1056/NEJMc2009316). Without knowing the prevalence of positivity in asymptomatic patients it is difficult to know whether testing all patients or all staff is likely to be a more effective measure.

Many thanks for highlighting this important point. We had already emphasised the issue with the lack of point-prevalence data from asymptomatic individuals in similar healthcare settings in our discussion, however we have now modified this point to read:

“This figure is difficult to contextualise due to paucity of point-prevalence data from asymptomatic individuals in similar healthcare settings or the local community.” For contrast, we discuss a publication examining prevalence of SARS-CoV2 infection in asymptomatic residents of a care home during an outbreak.

Furthermore, we have amended an additional sentence to read: “This may be partly explained by the relatively low rates of infection in the East of England in comparison to other areas of the UK (cumulative incidence 0.17%, thus far).”

With regards to prevalence in asymptomatic patients, this would be important information, however such data is not yet available.

7) Similarly more detail on infection control measures in your wards would be helpful: were FFP3 masks routinely fit tested, were they re-used, ventilation of wards (air changes per hour), use of high flow nasal oxygen and non-invasive ventilation, availability of single rooms and airborne infection isolation rooms, measures to limit staff-staff transmission? This is particularly important given your finding that the prevalence of positivity was higher in red zone wards.

Thank you for this suggestion. We have included a more detailed description of the hospital’s COVID-19 infection control and PPE practices as a caption to table S2 (formerly table s4) of the revised manuscript.

8) The discussion needs to be tightened up significantly so that it focuses on the findings of the study and does not overstate implications of those findings. Examples include: First two paragraphs of the discussion are not based on your findings or any data and are possibly specific to the UK. My hospital runs on-site PCR assays five times daily, without the need for a "substantial collaboration". It would be better to present your major findings first.

Thank you for this feedback. We have restructured the Discussion section, and removed UK-centric content. We have chosen to highlight the utility of collaborative efforts in enabling rapid development and upscale of a HCW screening programme in the context of limited pre-existing clinical laboratory capacity, as this is a challenge that we believe will be shared by many other institutions internationally.

9) "Our data clearly demonstrate that a relentless focus on the testing of individuals fitting a strict clinical case definition for COVID-19 will inevitably miss a substantial burden of asymptomatic and pauci-symptomatic disease". I would suggest that a prevalence of <1% (after you exclude previously symptomatic staff) is better described as small but possibly important, rather than substantial.

We have removed the words “a substantial burden of” a suggested.

10) "The value of this approach is illustrated by our detection of a cluster of cases in ward F, where the consequences of uncontrolled staff-to-patient transmission had the potential to cause substantial morbidity and possible mortality". I could not find any data demonstrating staff-to-patient transmission.

Thank you for questioning this. We have now re-worded this section of the discussion to improve clarity, as follows “The value of this approach is illustrated by our detection of a cluster of cases in ward F, where the potential for uncontrolled staff-to-staff or staff-to-patient transmission could have led to substantial morbidity and mortality in a particularly vulnerable patient group”.

11) “Coronaviruses generally have very low mutation rates (~10-6 per site per cycle)28, with the first reported sequence of the current pandemic only published on 12th January 2020. Whilst viral genomics is a powerful tool in the investigation of potential transmission events when combined with epidemiological data, given that SARS CoV-2 was only introduced into the human population in late 2019, there is at present a lack of diversity in circulating strains. As the pandemic unfolds and more detailed epidemiological and genome sequence data from patient and HCW clusters are generated, real-time study of transmission dynamics will become an increasingly important means of informing disease control responses. Prospective epidemiological data will also inform whether hospital staff are more likely infected in the community or at work, and may identify risk factors for the acquisition of infection, such as congregation in communal staff areas or inadequate access to PPE".This seems a little speculative and I can't really see the relevance to your findings. You argue that in the face of falling community transmission screening of HCWs becomes more important. If community infection is uncommon then the most likely source of transmission to a HCW is a patient. If so, then doesn't screening of all patients make more sense? This is particularly so as, even when community transmission is relatively high (based on the high deaths per capita in UK), the prevalance is approximately 0.6% after you exclude HCWs who were previously symptomatic.

The reviewer is correct in that as cases drop, the ability to identify hospital acquired infections rapidly will be critical to ensuring infection control measure are effective and that to achieve this testing of both HCW and patients will be key. Our intention was to highlight that HCW screening should be implemented as part of a package, which would of course include patient screening. We have updated the text in the discussion as follows:

“However, as the pandemic unfolds and detailed epidemiological and genome sequence data from patient and HCW clusters are generated, real-time study of transmission dynamics will become an increasingly important means of informing disease control responses and rapidly confirming (or refuting) hospital acquired infection. Importantly, implementation of such a programme would require active screening and rapid sequencing of positive cases in both the HCW and patient populations. Prospective epidemiological data will also inform whether hospital staff are more likely to be infected in the community or at work, and may identify risk factors for the acquisition of infection, such as congregation in communal staff areas or inadequate access to PPE.”

12) A lack of data on staff behaviour outside of work should be given as a weakness of the study.

We have added to the limitations section of our discussion: “Our study is limited by the relatively short time-frame, and small number of positive tests *and a lack of behavioural data.*”

13) Minor comments:

Discussion: Parameterise not parametrise

Abstract, findings. Use of the word "conversely" implies to me that being symptoms >7 days prior to testing and being truly asymptomatic or pauci-symptomatic were mutually exclusive categories. Presumably they are not.

I didn't find the vignettes to be at all useful.

As requested by reviewer 2, we have updated the vignettes to add more detail. We have removed the word “conversely”. (See Table 2.)

Reviewer #6:This manuscript report the results of COVID-19 screening study among healthcare workers. The study was conducted between 6th and 24th of April 2020 and 10,32 asymptomatic HCWs were screened in a 1300-bed teaching hospital, targeting wards considered to be at higher risk of transmission. There were 31 asymptomatic cases during the study period of which 12 previously experienced symptoms compatible with COVID-19 and 55% of the cases were truly asymptomatic. The authors may wish to consider the following comments:1) Abstract: I would state in the Methods section whether each HCW was tested once or more during the three week period. In the Findings section I would report n(%) to avoid confusion. I assume the 75% refer to 75% of the 12 HCWs (38.7%) who experiences symptoms.

Many thanks for this suggestion. We have added details as suggested to the Findings section. 21 individuals underwent repeat testing for a variety of reasons, discussed in the first paragraph of the Results section. We have not added these details to the Methods section of the Abstract due to space constraints.

2) The Methods section should provide more details about the study design and procedures with the full description in the appendix. For example, how many tests were done by HCW? Which departments/clinical areas were included in the study? How the data about symptoms were collected? There is a mention of a telephone consultation, but was this done on the day of the test, was there a follow-up? A paragraph providing a summary of these issues in the Methods section would be helpful so that the reader would not need to read the appendix to understand the study.

Thank you for this suggestion. We have added content to the first paragraph of the methods section to further explain these aspects of our study design. The number of HCWs or household contacts that underwent repeat testing is included in the first paragraph of the Results section (See Methods and Results sections)

3) What are the "defined groups" that are mentioned in the data analysis section in relation to Fisher's exact test? Similarly, what are the "categories" that are mentioned in relation to Man-Whitney test? Also, what was/were the variable(s) tested using Man-Whitney test? This section should be more specific to the present study and currently it is very generic and the same text could be used in any study.

We apologise for the lack of clarity regarding the statistical tests applied. In line with the suggestion made by the reviewer, we have adjusted the methods section accordingly and added clarity as to which tests have been applied and where.

4) Figure 1: I would report the median and interquartile range in each group, in the figure legend. It is suggested that the 95% CIs of the median were calculated, how were they calculated? It is stated "…viral loads were significantly lower for those in the HCW asymptomatic screening group than in those tested due to the presence of symptoms (Figure 1)". Looking at Figure 1, the data suggest that the median in the asymptomatic group is slightly above 30 while the median in the symptomatic group is approximately below 25. I am not sure if I missed something here.

We apologise to the reviewer for this error in the figure legend. The figure does indeed show the median and interquartile range, as the reviewer suggests it should. We have corrected the legend accordingly.

In terms of the relationship between CT value and symptom group, it should be noted that there is an inverse relationship between CT value and viral load (i.e. individuals with higher viral loads will have the presence of virus detected at an earlier cycle in the RT-PCR process and hence have lower CT values). We note that further clarity on this point was also requested by reviewer 4. We have amended the figure legend and the methods to make this clearer (see Figure 1).

5) Figure 3: Were "% of total positives" calculated for total population or within green, amber and red separately?

The % of total positives refers to all positive test results in total population, and not treating green/amber/red separately. We have amended the figure legend for clarity (Figure 1—figure supplement 1).

6) Figure 1—figure supplement 1: This legend of this graph mentions linear regression, and this should be stated in the statistical analysis section. Also, if linear regression were used, why are the data points being reported as median rather than mean? Were there concerns about whether the variable is approximately normally distributed? If yes, how was this handled in the linear regression considering that the assumptions of the linear regression may have been violated?

We thank the reviewer for this observation which is in line with those made by reviewers 2 and 4, and already addressed above. In brief, we agree that linear regression was not the appropriate statistical method given that normality cannot be assumed for these data. We have replaced Figure 1—figure supplement 1 with a non-parametric analysis. This analysis does not change our findings (of a lack of relationship between Ct value and symptom duration). (See Figure 1—figure supplement 1.)

7) I am not sure how informative the p-values from Fisher's exact tests are? The number of positive cases are small and using descriptive statistics may be more appropriate. I suggest that all the comparisons done between cases and non-cases could be presented in on table. This could be either n(%) for binary/categorical variables (such as symptomatic vs asymptomatic and red vs green vs amber etc.) and median (IQR) for continuous non-parametric variables (such as viral load). Such a table would provide a clear summary of the main results and conclusions of the study in one place.

A nice feature of Fisher's exact test is that it can be used with small sample sizes because it does not rely on asymptotic approximations. See, for instance, the “Handbook of Biological Statistics” by John H. McDonald, available at http://www.biostathandbook.com/fishers.html. From the Summary: “Use the Fisher's exact test of independence when you have two nominal variables and you want to see whether the proportions of one variable are different depending on the value of the other variable. Use it when the sample size is small.” For this reason, we have elected to maintain the use of Fisher’s exact test in Results section, but thank the reviewer for their suggestion.

Reviewer #7:I have to congratulate Rivett et al. for their great work in providing much need justification for testing of healthcare workers during this COVID-19. I just have a few minor comments.

Very many thanks for the complimentary assessment of our work.

Minor Comments:1) Is there any available data from the study to address the following questions, as they may provide useful insights to the transmission dynamics:a) Were there any cases that were tested negative initially, but developed COVID-19 symptoms and were tested positive later?

No HCWs in the symptomatic arm had previously tested negative, likely because prior to the inception of the screening programme none had been screened. The period of screening examined reflected the first three weeks of the programme. We anticipate seeing such individuals as the programme develops.

2) Are HCWs from symptomatic household contacts tested? If so, how many positive HCWs cases are link to household contacts?

HCWs with symptomatic household contacts could be tested simultaneously if requested. In all four of the cases where household contacts tested positive, the HCW who lived with them also tested positive.

3) Did any family members of HCWs with positive tests from the asymptomatic group subsequently develop COVID-19?

Thank you for this question, this information was not collected.

4) It is interesting to note that in your study, appropriately 32% of HCWs with positive tests from the asymptomatic group in this study, had experience symptoms compatible with COVID-19 > 7 days prior to testing. 75% of those had returned to work after 7 days self-isolation according to UK recommendations. Does this strengthen the need to test HCWs before they return to work?

This may strengthen the need to test HCWs prior to their return to work, however given the evidence from Wolfel et al., 2020, it seems that viable virus is not culturable beyond 8 days of symptom onset. Admittedly this was a small study involving healthy participants with mild disease. This would suggest that the current 7 days of isolation after onset of symptoms is relatively safe from an infection control standpoint. On the basis of this evidence, we believe that we are likely to be detecting prolonged shedding of non-viable viral RNA. More studies are clearly required to elucidate how long viable virus can actually be shed to better inform screening/return to work policies. This is now fully considered in the Discussion.